# Towards Better Understanding of In-Context Learning Ability from In-Context Uncertainty Quantification

**Shang Liu** *s.liu21@imperial.ac.uk*
*Imperial College Business School*
*Imperial College London*

**Zhongze Cai** *z.cai22@imperial.ac.uk*
*Imperial College Business School*
*Imperial College London*

**Guanting Chen** *guanting@unc.edu*
*Department of Statistics and Operations Research*
*University of North Carolina*

**Xiaocheng Li** *xiaocheng.li@imperial.ac.uk*
*Imperial College Business School*
*Imperial College London*

**Reviewed on OpenReview:** *https://openreview.net/forum?id=Jwtpbhheoy*

## Abstract

Predicting simple function classes has been widely used as a testbed for developing theory and understanding of the trained Transformer's in-context learning (ICL) ability. In this paper, we revisit the training of Transformers on linear regression tasks, and different from all the existing literature, we consider a bi-objective prediction task of predicting both the conditional expectation $\mathbb{E}[Y|X]$ and the conditional variance $\text{Var}(Y|X)$. This additional uncertainty quantification objective provides a handle to (i) better design out-of-distribution experiments to distinguish ICL from in-weight learning (IWL) and (ii) make a better separation between the algorithms with and without using the prior information of the training distribution. Theoretically, we show that the trained Transformer reaches near Bayes optimum, suggesting the usage of the information of the training distribution. Our method can be extended to other cases. Specifically, with the Transformer's context window $S$, we prove a generalization bound of $\tilde{\mathcal{O}}(\sqrt{\min\{S,T\}/(nT)})$ on $n$ tasks with sequences of length $T$, providing sharper analysis compared to previous results of $\tilde{\mathcal{O}}(\sqrt{1/n})$. Empirically, we illustrate that while the trained Transformer behaves as the Bayes-optimal solution as a natural consequence of supervised training in distribution, it does not necessarily perform a Bayesian inference when facing task shifts, in contrast to the *equivalence* between these two proposed in many existing literature. We also demonstrate the trained Transformer's ICL ability over covariate shift and prompt-length shift and interpret them as a generalization over a meta distribution.

## 1 Introduction

A particularly remarkable characteristic of Large Language Models (LLMs) is their ability to perform in-context learning (ICL) (Brown et al., 2020). Once pretrained on a vast corpus of data, LLMs can solve newly encountered tasks when provided with just a few training examples, without any updates to LLMs' parameters. ICL has significantly advanced the technique known as prompt engineering (Ekin, 2023), which has achieved widespread success in various aspects of daily life (Oppenlaender et al., 2023; Heston & Khun, 2023; Li et al., 2023a). Behind the empirical success of ICL, this method has captured the attention of the

theoretical machine learning community, leading to considerable efforts into understanding ICL from different theoretical perspectives (Xie et al., 2021; Akyürek et al., 2022; Von Oswald et al., 2023; Zhang et al., 2023a).

This work aims to enhance the theoretical understanding of ICL by examining the Transformer's context window and showing its effects on the approximation-estimation tradeoff. Although we obtain the results for the case of uncertainty quantification where the model is asked to predict both the mean value and the uncertainty of its prediction, our analysis is applicable across various ICL tasks and provides sharper bounds compared to previous works. In addition to developing theories, we empirically demonstrate the effectiveness of Transformers to in-context predicting the mean and quantifying the variance of regression tasks. We design a series of out-of-distribution (OOD) experiments, which have generated significant interest within the community (Garg et al. (2022); Raventós et al. (2024); Singh et al. (2024)). These experiments provide insights in designing the pretraining process and understanding the ICL capabilities of transformers.

Our contributions are as follows:

- We theoretically analyze the problem of in-context uncertainty quantification. We consider the case when Transformers can only process the contexts within a context window capacity $S$ and derive a generalization bound of $\tilde{\mathcal{O}}(\sqrt{\min\{S, T\}/nT})$ for pretraining over $n$ tasks with sequences of length $T$ (Theorem 3.2). Our result can be easily extended to other cases under the assumption of almost surely bounded and Lipschitz loss functions. As far as we know, our generalization bound is the first of its kind and provides a tighter bound compared to the existing analyses (Li et al., 2023b; Zhang et al., 2023b) when $S < T$. In particular, we use the context-window structure to establish a Markov chain over the prompt sequence and construct an upper bound for its mixing time. We also examine the extra approximation error term due to a finite context window $S$ (Section B.2). Combining those discussions together, we quantify the convergence of the trained Transformer's risk to the *Bayes-optimal* risk. Moreover, we note that all the theoretical results only show that the trained Transformer achieves a near-optimal in-distribution risk compared to that of the Bayes-optimal predictor. It is incorrect to draw (from the theory or the in-distribution numerical results) either of the conclusions that (i) the Transformer that achieves the near-optimal risk exhibits a similar structure as the Bayes-optimal predictor by performing Bayesian inference (Zhang et al., 2023b; Panwar et al., 2023) or (ii) the Transformer performs as the Bayes-optimal predictor for out-of-distribution tasks.

- Numerically, for the uncertainty quantification problem, we provide a comprehensive study of the in-context learning ability of the trained Transformer under three scenarios of distribution shifts: task shift (Section 4.1), covariate shift (Section 4.2), and prompt length shift (Section 4.3). We find that transformers are capable of in-context learning of both mean and uncertainty predictions, even under a moderate amount of task distribution shift, provided that the task diversity in the training data is relatively large. Additionally, we find that increasing the task diversity with a meta-learning approach helps the transformer learn in-context robustly under covariate shift. Lastly, we observe that removing positional encoding from the embedding vector massively helps the generalization ability, enabling it to better learn tasks in-context with unseen prompt length.

We defer more discussions on the related literature to Section A.

## 2 Problem Setup

Consider training a Transformer for some regression task $f: \mathcal{X} \to \mathcal{Y}$ from a function class $\mathcal{F}$. The covariates $x \in \mathcal{X} \subset \mathbb{R}^d$ are generated from a distribution $\mathcal{P}_{\mathcal{X}}$, and the output variable $y = f(x) + \sigma \cdot \epsilon$ for some function $f \in \mathcal{F}$, noise level $\sigma$, and some random noise $\epsilon$ with $\mathbb{E}[\epsilon] = 0$ and $\text{Var}(\epsilon) = 1$. The Transformer performs a sequential prediction task over the following sequence

$$(x_1, y_1, ..., x_T, y_T)$$

where $T$ is the total number of (in-context) samples. For a Transformer model with parameters $\theta \in \Theta$, we denote it as $\texttt{TF}_\theta$. At each time $t = 1, ..., T$, the model $\texttt{TF}_\theta$ observes $H_t := (x_1, y_1, ..., x_{t-1}, y_{t-1}, x_t)$ (which is called *history* or *prompt*) and makes a bi-objective prediction of $y_t$ to both predict the mean with $\hat{y}_\theta(H_t)$ and quantify the uncertainty of the prediction with $\hat{\sigma}_\theta(H_t)$. With a slight abuse of notations, we denote the

output of the model by $\mathrm{TF}_\theta(H_t) := (\hat{y}_\theta(H_t), \hat{\sigma}_\theta(H_t))$. The pretraining dataset consists of $n$ sample sequences

$$\mathcal{D} := \left\{ \left( x_1^{(i)}, y_1^{(i)}, x_2^{(i)}, y_2^{(i)}, ..., x_T^{(i)}, y_T^{(i)} \right) \right\}_{i=1}^n .$$

To generate each sample sequence in $\mathcal{D}$, a function $f_i$ is sampled from a distribution $\mathcal{P}_\mathcal{F}$ supported on $\mathcal{F}$ and a noise level $\sigma_i$ is sampled from a distribution $\mathcal{P}_\sigma$ supported on $[0, \bar{\sigma}] \subset \mathbb{R}$. Then each $x_t^{(i)}$ and $y_t^{(i)}$ is generated pairwise by

$$x_t^{(i)} \overset{\text{i.i.d.}}{\sim} \mathcal{P}_\mathcal{X}, \quad y_t^{(i)} = f_i\left(x_t^{(i)}\right) + \sigma_i \cdot \epsilon_t^{(i)}, \quad \epsilon_t^{(i)} \overset{\text{i.i.d.}}{\sim} \mathcal{P}_\epsilon$$

where $\epsilon_t^{(i)}$'s are i.i.d. noise of mean zero and unit variance.

The Transformer is trained by minimizing the following empirical loss

$$\hat{\theta}^{\mathrm{ERM}} := \arg\min_{\theta \in \Theta} \frac{1}{nT} \sum_{i=1}^n \sum_{t=1}^T \ell\left( \mathrm{TF}_\theta\left(H_t^{(i)}\right), y_t^{(i)} \right) \tag{1}$$

where $H_t^{(i)} = (x_1^{(i)}, y_1^{(i)}, ..., x_{t-1}^{(i)}, y_{t-1}^{(i)}, x_t^{(i)})$ and $l((\cdot, \cdot), \cdot) : (\mathbb{R} \times \mathbb{R}^+) \times \mathbb{R} \to \mathbb{R}$ denotes the loss function. We use $x_t^{(i)}, y_t^{(i)}, H_t^{(i)}$ to denote the samples in the training dataset and $x_t, y_t, H_t$ to denote an arbitrary feature, label, and history. Throughout this paper, we assume that each probability distribution is continuous and has a probability density function (p.d.f.), and we also assume the conditional distribution of $y_t$ on observing $H_t$ exists almost surely.

The loss function is accordingly defined by

$$\ell\left((\hat{y}, \hat{\sigma}), y\right) := \log \hat{\sigma} + \frac{(y - \hat{y})^2}{2\hat{\sigma}^2}.$$

**Definition 2.1** (Bayes-optimal predictor). The Bayes-optimal predictor under the distributions $\mathcal{P}_\mathcal{F}$, $\mathcal{P}_\mathcal{X}$, $\mathcal{P}_\sigma$ and $\mathcal{P}_\epsilon$ is defined by

$$(y_t^*(\cdot), \sigma_t^*(\cdot)) \in \arg\min_{(y(\cdot), \sigma(\cdot)) \in \mathcal{G}_t \times \mathcal{G}_t} \mathbb{E}\left[ \ell\left( \left(y(H_t), \sigma(H_t)\right), y_t \right) \right] \tag{2}$$

where $\mathcal{G}_t$ is the class of all measurable functions of $H_t \in \mathcal{H}_t$. The expectation is taken with respect to the following dynamics: $x_t \sim \mathcal{P}_\mathcal{X}$, $\epsilon_t \sim \mathcal{P}_\epsilon$, $f \sim \mathcal{P}_\mathcal{F}$, $\sigma \sim \mathcal{P}_\sigma$, $y_t = f(x_t) + \sigma \cdot \epsilon_t$ and $H_t = (x_1, y_1, \ldots, x_t)$.

The loss on the right-hand-side of equation 2 is the expectation of the empirical loss equation 1. With a rich enough function class and an infinite amount of training samples, the trained Transformer $\mathrm{TF}_{\hat{\theta}_{\mathrm{ERM}}}$ converges to $(y_t^*, \sigma_t^*)$ as will be shown in Theorem 3.2.

## 2.1 Motivation for the uncertainty quantification objective

We first give a semi-formal definition for in-context learning and in-weight learning of the bi-objective linear regression task considered in this paper.

*In-context learning* refers to that the Transformer gains the ability to learn from the in-context samples (samples in $H_t$) and behave as an algorithm. For example, the Transformer exhibits in-context learning when it behaves like a ridge regression model when performing the linear regression task. The in-context learning ability should persist under the out-of-distribution setting such as shifting $\mathcal{P}_\mathcal{X}$, $\mathcal{P}_\mathcal{F}$, and $\mathcal{P}_\sigma$.

*In-weight learning*, for the NLP tasks, generally refers to that the Transformer relies on the information stored in its weights to make predictions, rather than the in-context samples. In this light, it memorizes the training samples and uses the memorization to make predictions. This memorization mechanism is more aligned with supervised learning setting; in particular, the mechanism is sensitive to distribution shifts and thus is not considered a desirable outcome of training Transformers.

Then the question is when we train the Transformer according to the objective equation 1, does it exhibit in-context learning or in-weight learning? For the classic single-objective linear regression task, we note two

facts: (a) the trained Transformer is near the Bayes-optimal predictor (Xie et al., 2021; Zhang et al., 2023b; Panwar et al., 2023); (b) under a Gaussian prior, the Bayes-optimal predictor is exactly a ridge regression model (with proper choice of the regularization parameter, (Wu et al., 2023)). With these two facts, it is tempting to draw the conclusion that the Transformer gains in-context learning ability and behaves as a ridge regression model, but this has to be done with caution or may even lead to a wrong conclusion, given that the trained Transformer does not exhibit out-of-distribution ability for full generality as noted in the numerical experiments (Garg et al., 2022). This motivates us to consider the bi-objective tasks. First, for the uncertainty quantification objective, when the number of in-context samples is fewer than the dimension of $x_t$'s, a near Bayes-optimal predictor must utilize the information in the training procedure, whereas there is no algorithm that can optimally do this via only in-context samples. Thus it well distinguishes the Bayes-optimal predictor from any possible algorithm and therefore gives a clearer picture of whether the trained Transformer indeed behaves as an algorithm. Second, the uncertainty quantification objective provides us an easy handle to designing numerical experiments, such as the *flipped* experiments (Wei et al., 2023; Singh et al., 2024), to distinguish in-context learning from in-weight learning. Third, the uncertainty quantification objective is of independent interest as it indicates whether the trained Transformer knows its uncertainty or not. Furthermore, through this bi-objective task, we illustrate that while the trained Transformer is near the Bayes-optimal predictor, it does not necessarily perform Bayesian inference when making predictions, which contradicts the arguments in (Zhang et al., 2023b; Panwar et al., 2023; Jeon et al., 2024).

## 3 In-Context Learning when In-Distribution

In this section, we focus on the in-distribution property of the trained Transformer. We provide a finite-sample analysis of how trained Transformers reach near Bayes-optimum. While our analysis is made on the case of uncertainty quantification, it can be easily adapted to other loss functions such as mean squared error. To proceed, we first provide the exact form of the Bayes-optimal predictor defined in equation 2 for the mean and uncertainty prediction.

**Proposition 3.1** (Bayes-optimal predictor for mean and uncertainty prediction)**.** *The Bayes-optimal predictor of the step-wise population risk defined in equation 2 is given by*

$$y_t^*(H_t) = \mathbb{E}[y_t|H_t], \quad \sigma_t^{*2}(H_t) = \mathbb{E}[(y_t - y_t^*(H_t))^2|H_t] = \mathbb{E}[(f(x_t) - y_t^*(H_t))^2|H_t] + \mathbb{E}[\sigma^2|H_t].$$

The optimal mean predictor shares the same form as the Bayes-optimal predictor for a single-objective mean prediction task. The additional uncertainty prediction task does not change the nature of the mean prediction part. The two terms in the optimal uncertainty predictor can be interpreted as follows. The first term is epistemic uncertainty, which indicates the uncertainty (of identifying the $f$ that governs the history $H_t$) due to lack of information. The term decreases as the samples accumulate, i.e., as the number of in-context samples $t$ increases. The second term is aleatoric uncertainty also known as intrinsic uncertainty.

Recall that the empirical risk estimator is defined by equation 1. Now we define the population risk as

$$R(\text{TF}_\theta) := \frac{1}{T}\mathbb{E}_{H_t}\left[\sum_{t=1}^{T}\ell\big(\text{TF}_\theta(H_t), y_t\big)\right],$$

where $H_t$ is another sampled sequence that is independently and identically distributed as $H_t^{(i)}$'s in the training data. We denote the population risk minimizer as $\theta^*$:

$$\theta^* \in \underset{\theta \in \Theta}{\arg\min}\, R(\text{TF}_\theta). \tag{3}$$

Now we present our main theoretical result.

**Theorem 3.2.** *Let $\hat{\theta}^{ERM}$ denote the ERM estimator as defined in equation 1 over the function class of the $L$-layer, $M$-heads Transformer models. Suppose that at each time $t$, the Transformer has a context window of making predictions based on $x_t$ and previous $S$ pairs of $(x_s, y_s)$ for $s = \max\{1, t - S\}, \ldots, t - 1$. Then under*

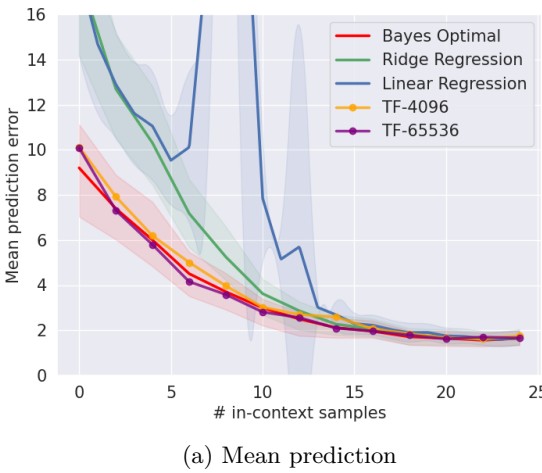
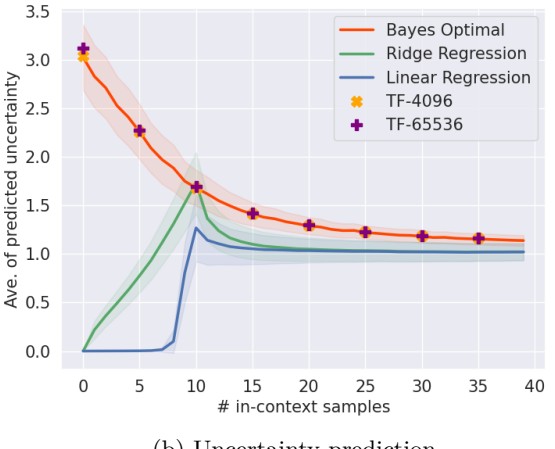

(a) Mean prediction

(b) Uncertainty prediction

Figure 1: Transformer behaves close to the Bayes-optimal predictor for in-distribution tasks. Details of the distributions in data generation are given in Section G.1. The numbers 4096 and 65536 refer to the number of tasks (configurations of $(w_i, \sigma_i)$) used in the training, which is formally defined in Section G.2. The Bayes-optimal predictor is stated in Proposition 3.1 and calculated analytically in Section G.3. For the left panel, the y-axis gives the mean squared error in predicting $y_t$. For the right panel, the $y$-axis gives the average of the predicted uncertainty over all the test samples (average of $\hat{\sigma}(H_t)$ or $\sigma^*(H_t)$ on test samples). In particular, we note that ridge regression and linear regression (ordinary least squares) do not naturally produce a measurement of uncertainty, so we use the sum of residuals on the in-context samples as their estimates of uncertainty. More visualizations are deferred to Section C.1.

*some boundedness assumptions of the Transformer's parameters (Assumption B.5 and B.6), we have with probability at least $1 - \delta$,*

$$R(TF_{\hat{\theta}^{ERM}}) - R(TF_{\theta^*}) \leq \tilde{\mathcal{O}} \left( \sqrt{\min\{S, T\}/(nT)} \right).$$

*where $\tilde{\mathcal{O}}$ omits poly-logarithmic terms that depend on $n, T, 1/\delta$ and boundedness parameters.*

**Proof sketch.** First, we prove that (a slightly redefined version of) the truncated history forms up a Markov chain conditioned on observing the full hidden information $f^{(i)}$ and $\sigma^{(i)}$, and upper bound the mixing time by $\min\{S, T\}$ to enable the concentration arguments. Second, we prove that the loss function is almost surely bounded (Lemma E.3) in preparation for McDiarmid-type concentration inequalities (Lemma F.2, (Paulin, 2015)). Third, we show that the loss is almost surely Lipschitz to control the difference between loss functions with respect to the change of the parameter (Lemma E.7). Fourth, we prove that there exist two distributions $\rho_{\hat{\theta}^{ERM}}$ and $\pi$ over parameter space $\Theta$, satisfying a number of properties as constructed in Lemma E.11. Lastly, we use standard PAC-Bayes arguments over $\rho_{\hat{\theta}^{ERM}}$ and $\pi$ and conclude the proof. The detailed proofs are deferred to Section D.2.

**Comparison with previous results.** There are also other theoretical results that characterize the outcomes of the (pre-)training on Transformer models (Zhang et al., 2023a; Wu et al., 2023; Xie et al., 2021; Li et al., 2023b; Bai et al., 2024; Zhang et al., 2023b; Lin et al., 2023a). Our analysis differs from theirs in terms of both the conclusion and the techniques. One stream of results examines the property of the gradient flow (or gradient descent) over the loss function for linear regression problems. The exact quantification of the gradient flow entails a simplification of the Transformer's architecture to the case of a single-layer attention mechanism under linear activation or even simpler settings (Zhang et al., 2023a; Wu et al., 2023). While their analyses provide insights into the learning dynamics of Transformer models, the learning of the single-layer attention Transformer can be very different from multiple-layer Transformers (Olsson et al., 2022; Reddy, 2023). Another major line of research uses statistical learning arguments (Xie et al., 2021; Li et al., 2023b; Bai et al., 2024; Zhang et al., 2023b; Lin et al., 2023a) such as algorithm stability, chaining, or PAC-Bayes arguments. Bai et al. (2024) focus on making predictions after observing a fixed length of variables under the i.i.d. setting (which is more aligned with the standard supervised learning setting), which differs from the

more practical setting of making predictions at every position as in Theorem 3.2. Xie et al. (2021) prove the convergence between the Bayesian inference and the true underlying distribution rather than the trained model and the Bayesian inference. Lin et al. (2023a) consider a sequential decision-making problem and use covering arguments to derive generalization bounds, while their analysis does not adopt the concentration arguments inside each sequence, resulting in an $\tilde{\mathcal{O}}(\sqrt{1/n})$ upper bound for the average regret. The most related works to ours are Li et al. (2023b); Zhang et al. (2023b). The major difference is that they all consider the only case of $S \geq T$. Li et al. (2023b) use the algorithm stability arguments to give a generalization bound over $|R - r|$ of order $\tilde{O}(\sqrt{1/(nT)})$. They prove the loss difference caused by perturbing one input pair over a history of length $t$ is controlled by $\mathcal{O}(1/t)$. Averaging those differences leads to a $\mathcal{O}(\log(T)/T) = \tilde{\mathcal{O}}(1/T)$ inside each sequence (see their equation (15) in their Appendix C), which appears in the Azuma-Hoeffding argument to prove that the loss per sequence is $\tilde{\mathcal{O}}(T^{-1/2})$-sub-Gaussian. However, in the case of $S \ll T$ (which is more often the case in practice), the algorithm stability term is of $\mathcal{O}(1/S)$. Averaging these terms inside each sequence leads to a difference of order $\mathcal{O}(1/S)$. If we stick to the original Azuma-Hoeffding arguments, the sum of squares of these terms is of $\mathcal{O}(T/S^2)$, leading to a far worse sub-Gaussian norm of $\mathcal{O}(T^{1/2}S^{-1})$, resulting in a final generalization bound of order $\tilde{O}(T^{1/2}S^{-1}n^{-1/2})$ that is clearly suboptimal compared to our $\tilde{O}(\sqrt{S/(nT)})$. Besides, such a bound also grows with $T$, which is undesirable. Similar to ours, Zhang et al. (2023b) also use a concentration argument for Markov chains. However, their Theorem 5.3 has two limitations: The first is that their result is of the order $\tilde{\mathcal{O}}(\sqrt{\tau_{\min}/(nT)})$ but they do not specify $\tau_{\min}$. Since they do not consider the truncated history but the full history, the Markov chain (which is not verified by them) will never mix inside each task sequence (see our discussions in Section D.2). Thus, the term $\tau_{\min}$ in their result is actually $T$, leading to an order of $\tilde{\mathcal{O}}(\sqrt{1/n})$, which is suboptimal compared to our $\tilde{\mathcal{O}}(\sqrt{S/(nT)})$ when the context window $S \ll T$. The second limitation is that their error decomposition is not tight: their excessive risk bound (measured by the total variation distance between the distribution induced by $\hat{\theta}$ and that by $\theta^*$) has a term $D_{\mathrm{kl}}(\mathcal{P}_{\mathrm{true}}, \mathcal{P}_{\theta^*}) - \mathrm{TV}(\mathcal{P}_{\mathrm{true}}, \mathcal{P}_{\theta^*})$, which means their result has an extra term of the approximation error since the Kullback-Leibler divergence is stronger than the total variation distance (Polyanskiy & Wu, 2024). Our work is the first theoretical analysis showing the effects of the context window $S$ on the performance of the Transformer up to our knowledge. The construction of the truncated history serves two-fold: not only does the truncation fit the practical model of finite context window but it also gives an upper bound on the mixing time. Concentration inside each sequence makes it possible to analyze the training dynamics broader than fixed-length sequences and prove the convergence to near Bayes-optimum. The context window $S$ also captures a novel dimension of the approximation-estimation tradeoff in the Transformer model.

**Extension of Theorem 3.2 to other problems.** We remark that the result and its derivation do not pertain to the uncertainty quantification setting, but hold for more general loss functions and are of independent interests. In particular, our analysis still holds as long as the loss function is almost surely bounded and Lipschitz with respect to the change of parameter $\theta$, as we can see from the proof sketch. We note here that to enable the Markov chain's concentration arguments, the almost surely bounded loss requirement cannot be relaxed to other tail properties such as sub-Gaussian (see the counter example in Theorem 4 of Fan et al. (2021)).

We defer discussions on the approximation error to Section B.2.

## 4   In-Context Learning under Distribution Shifts

In Section 2, we describe in-context learning ability as algorithm-like that predicts based on the learning from in-context samples, and such an ability should be generalizable to an out-of-distribution (OOD) environment. In this section, we differentiate the OOD scenarios into task shift, covariate shift and length shift, and examine the Transformer's in-context learning ability in each scenario. As far as we know, we provide the first comprehensive group of numerical experiments (for the linear regression task) that demonstrates the Transformer's ability to handle these three types of distribution shifts. We provide preliminary theoretical discussions for such abilities and hope this points directions for future theoretical research.

### 4.1 Task shift

When the trained Transformer performs well on the OOD data, it means that the Transformer gains an algorithmic ability that learns to make predictions based on the in-context samples, because such an ability is not restricted to the distribution of the inputs. Comparatively, the mere observation that the Transformer works well on the in-distribution data does not demonstrate its in-context learning ability as a traditional supervised learning model also has such ability and generalization performance over in-distribution data.

In the previous section, when we show the in-distribution performance of the Transformer, the variance parameter $\sigma^2$ is generated by the prior of the inverse-Gamma distribution $\sigma^2 \sim \text{Inv-Gamma}(\underline{\tau}, \bar{\tau})$ with parameters $\underline{\tau}$ and $\bar{\tau}$. The details of the other generation distributions are deferred to Section G.1. For the in-distribution setting, we set $\underline{\tau} = \bar{\tau} = 20$ which leads to a prior mean around 1. Now we consider three out-of-distribution (OOD) settings for the

- S-OOD (small OOD): $\underline{\tau} = 80, \bar{\tau} = 20$. The prior mean of $\sigma$ is around 0.5.

- M-OOD (medium OOD): $\underline{\tau} = 100, \bar{\tau} = 400$. The prior mean of $\sigma$ is around 2.

- L-OOD (large OOD): $\underline{\tau} = 100, \bar{\tau} = 1600$. The prior mean of $\sigma$ is around 4.

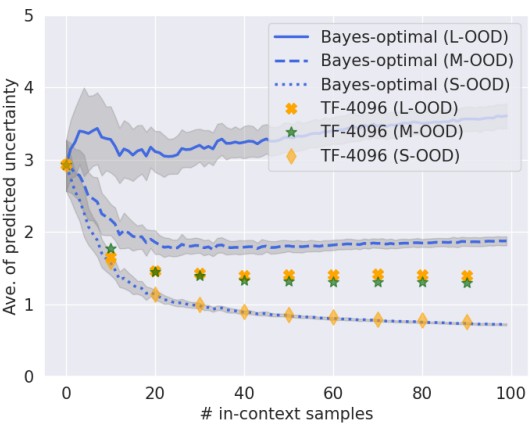 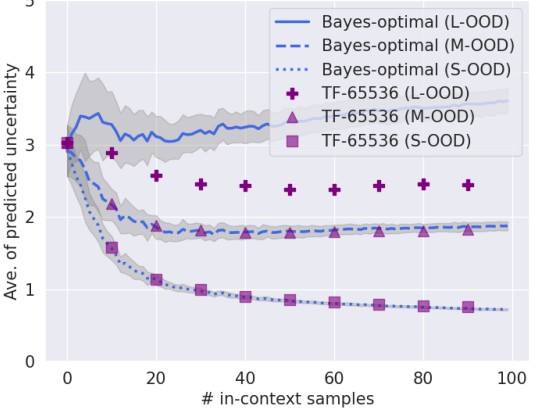

(a) Transformer trained w/ small pool size      (b) Transformer trained w/ large pool size

Figure 2: OOD performances of Transformers and the Bayes-optimal predictor. The $y$-axis gives the average of the predicted uncertainty over all the test samples (average of $\hat{\sigma}(H_t)$ or $\sigma^*(H_t)$ on test samples), and ideally, they should converge to the expected uncertainty level of 0.5 (S-OOD), 2 (M-OOD), and 4 (L-OOD) as in-context samples increase. There are three OOD environments: small (S-OOD), medium (M-OOD), and large (L-OOD) that reflect the intensity of the OOD. Two versions of the Transformer model are trained with a pool size of 4096 and 65536. The Transformers and the Bayes-optimal predictor are the same as the ones in Figure 1. The only difference is that they are evaluated on OOD data here.

We make following observations based on Figure 2: First, the Bayes-optimal predictor predicts well. We note that the Bayes-optimal is computed based on the in-distribution prior distribution (with respect to $\sigma^2$). Thus when the Bayes-optimal predictor is tested under the OOD environment as in Figure 2, the prior used by the Bayes-optimal predictor is wrong. But we note from Figure 2 that the Bayes-optimal predictor has the OOD ability to correct the prior as the in-context samples accumulate (noting that the three Bayes-optimal curves converging to the correct mean of 0.5, 2, and 4). This is also known as the washing out of priors in Bayesian statistics. Second, Transformers deviate from the Bayes-optimal on these OOD tasks. For both plots in Figure 2, we note that the predicted values from the Transformers deviate from those of the Bayes-optimal predictor when the OOD intensity is large. This tells that the trained Transformer does not conduct Bayesian inference under task shift. In other words, it is incorrect to conclude that the trained Transformer behaves as the Bayes-optimal predictor just from the matching in-distribution loss (as Figure 1). Moreover, the

Transformer achieves a near-optimal loss for in-distribution tasks (as Figure 1) but it does so via a different avenue than the Bayes-optimal predictor (as Figure 2). This is in contrast with the findings/claims in the previous papers (Zhang et al., 2023b; Panwar et al., 2023). Third, the deviation of the trained Transformer from the Bayes-optimal is smaller when the task diversity is large or the OOD intensity is small. This is aligned with the findings in (Raventós et al., 2024) for in-distribution performance, while the OOD setting is not studied therein.

The theoretical evidence only states that the trained Transformer has a near-optimal in-distribution loss as the Bayes-optimal predictor. But it does not give any evidence that these two have a structural similarity that persists for OOD tasks. In particular, we note that the trained Transformer may take *statistical shortcuts*: When evaluated under in-distribution tasks or some simple task shifts (e.g. scaling the weights vectors or changing the signal-noise ratio), Zhang et al. (2023a); Wu et al. (2023) show that Transformer will construct shortcuts using the statistical property of the training distribution. More specifically, Transformers (can, and will) encode the information of the covariance matrix into their model parameters to reach near-optimal in-distribution performance. Such statistical shortcuts are beneficial to the in-distribution performance but can hurt its OOD ability. Increasing the training task diversity, such as a larger training pool size, may remove some of these statistical shortcuts to obtain near-optimal empirical loss, and thus better enable its in-context learning ability.

We defer more discussions and visualizations on this OOD experiment to Section C.2.

### 4.2 Covariate shift

For all the numerical experiments so far, the covariates are generated from $\mathcal{N}(0, I_d)$. This follows the standard setup of the existing literature (Akyürek et al., 2022; Von Oswald et al., 2023; Li et al., 2023b; Raventós et al., 2024). It is also noted from the literature (Garg et al., 2022; Zhang et al., 2023a) that the trained Transformer in this way lacks in-context learning ability under covariate shift. However, if the Transformers trained on different tasks (different $f_i$'s) can generalize to unseen tasks during the test phase, one may wonder if the Transformers trained on different *covariates* (different distributions of $x_i$'s) can also generalize. In this subsection, we give a positive answer to that question by proposing a meta-training procedure that improves the trained Transformer's ability to handle covariate shifts. Specifically, we consider generating the covariates in the training data as follows:

- For each training sequence (say, the $i$-th), we first sample a vector $(\lambda_1, ..., \lambda_d)$ where each $\lambda_j$ is i.i.d. Uniform$[0, 2]$. Then all the $X_t^{(i)}$'s for $t = 1, ..., T$ are sampled from $\mathcal{N}(0, \text{diag}((\lambda_1, \ldots, \lambda_d)))$. In this sense, the covariance matrix of $X_t^{(i)}$'s is also a random variable, and the $X_t^{(i)}$'s can be viewed as being sampled in a hierarchical manner from a *meta-distribution.*

We examine the performance of such a training procedure under four OOD test settings. In other words, the $X_t^{(i)}$'s in the test data is generated from the following four distributions where $d = 8$.

- Large covariance (L-cov): $X_t^{(i)}$'s are sampled from $\mathcal{N}(0, 4I_d)$.

- Decreasing diagonal (Dec.): $X_t^{(i)}$'s are sampled from $\mathcal{N}(0, \text{diag}([d/i]_{i=1}^d))$.

- Shrinking diagonal (Shr.): $X_t^{(i)}$'s are sampled from $\mathcal{N}(0, \text{diag}([d/i^2]_{i=1}^d))$.

- Rotation (Rot.): $X_t^{(i)}$'s are sampled from $\mathcal{N}(0, U_i \text{diag}([d/i]_{i=1}^d) U_i^\top)$ where $U_i$ is an orthogonal matrix independently generated for each sequence.

Figure 3 gives the evaluation result under the 4 OOD settings. We note that the meta-distribution used is still significantly different from the four OOD test environments. Thus the results show the effectiveness of the meta-training approach. We evaluate the prediction error of models ordinarily trained, and models trained by the meta-training process. For both mean and uncertainty, the models trained by the meta-training procedure have a smaller prediction error.

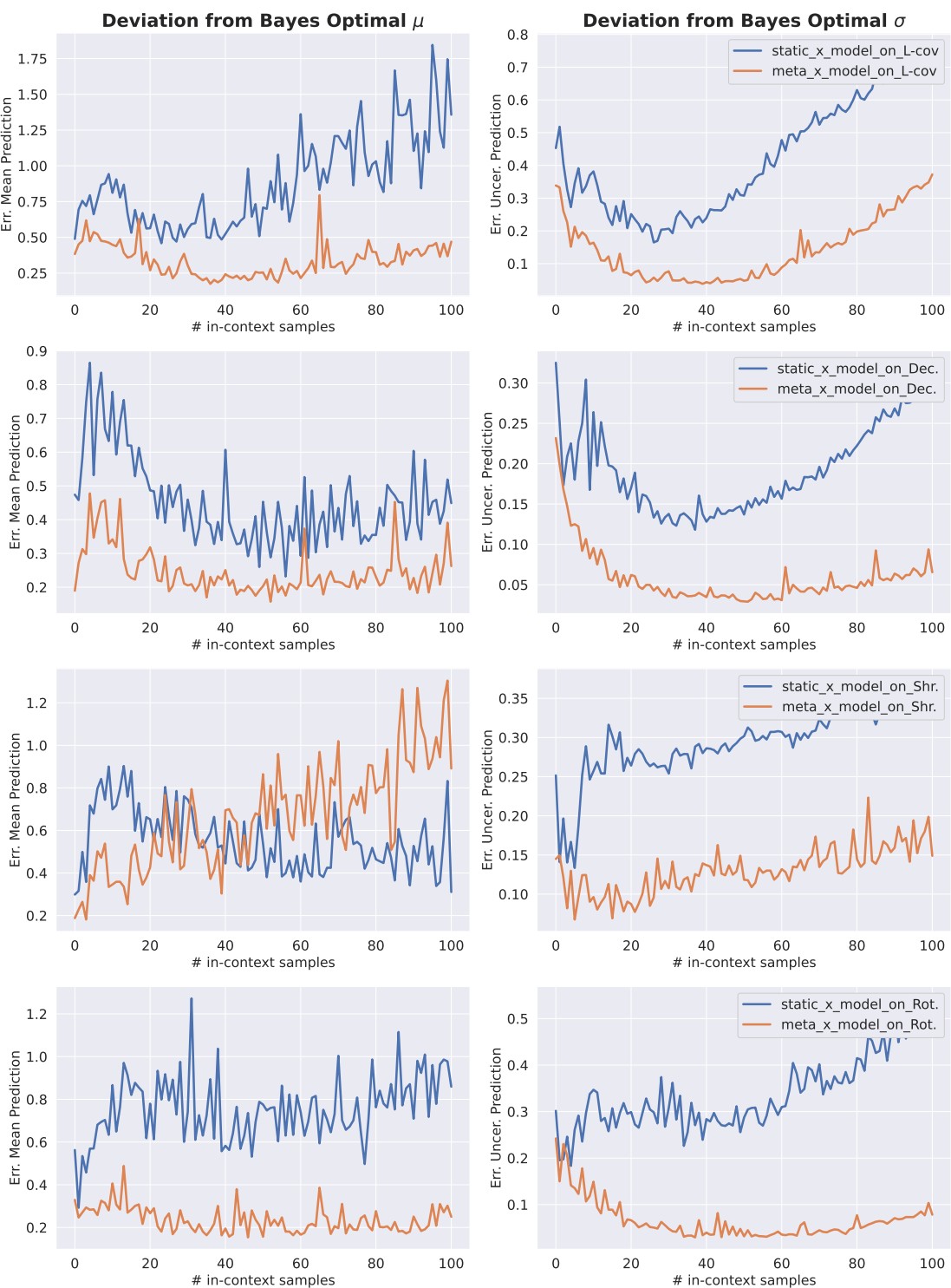

Figure 3: The errors of the mean and uncertainty prediction where the error is measured by the absolute difference against the Bayes-optimal predictor. The *static_x_model* corresponds to models trained with the standard way in generating $X_t$'s, while the *meta_x_model* corresponds to the new approach of drawing $X_t$'s from the meta-training procedure. In all 4 OOD settings, meta-trained models have better performance.

### 4.3 Length shift and positional embedding

Existing work (Dai et al., 2019; Anil et al., 2022; Zhang et al., 2023a) have pointed out the failure of Transformers to generalize to longer contexts than the ones they have seen during training. It is worth mentioning that the code implementations of some previous works (Zhang et al., 2023a; Garg et al., 2022) are based on the "transformers" package of Hugging Face. Although these works have not included positional embedding explicitly, the GPT2 module imported from this package adds a built-in positional encoding implicitly. We suspect that some unexpected behaviors (like the "unexpected spikes of prediction error" mentioned in Zhang et al. (2023a)) are due to that the built-in positional encoding is not disabled. In this subsection, we investigate the length generalization ability of the trained Transformer on the uncertainty quantification task. Specifically, we control the prompt lengths that the model is trained on. Previous experiments train the model on prompts with lengths (number of in-context samples) ranging from 1 to 100. In this experiment, we control the training prompts such that the lengths are either shorter than 44 or longer than 45 (the choice of 45 as the cutoff point is not essential). We specify these two configurations below.

- Trained on $\leq 44$: the model is trained on prompts with length ranging from 1 to 44, and is evaluated with prompt length from 1 to 100

- Trained on $\geq 45$: the model is trained on prompts with length ranging from 45 to 100, and is evaluated with prompt length from 1 to 100

We regard this difference in prompt length between training and testing as length shift. We evaluate the effect of removing positional encoding under this prompt length generalization task. If positional encoding is added to the embedding, samples at unseen positions will be associated with an unseen positional encoding vector in the embedding space. This requires the model to handle not just an unseen number of in-context samples, but also a possibly unseen embedding distribution, and generalization ability will likely deteriorate. As mentioned previously, the built-in positional encoding of GPT2 model use a positional encoding which is set to be $(t, 0, \cdots, 0)^\top$ for the $t$-th token, and the encoding will then be concatenated to the embedding vector. We validate the above intuitions with the following 4 training configurations.

- No positional encoding (w/o Pos.): the model is trained without positional encoding.

- Add positional encoding (w/ Pos.): the model is trained with GPT2's built-in positional encodings.

- Add segment encoding (w/ S-Pos.): the positional encoding is added with a random amount offset. For the $i$-th training sequence, a random offset $t_i$ is first uniformly sampled from $\{0, 1, \ldots, 22\}$. Next, for each token in this prompt at position $t$, the positional encoding is set to $(t + t_i, 0, \ldots, 0)^\top$.

- Add full range encoding (w/ F-Pos.): similar to the S-Pos. configuration, the positional encoding is added with a random amount offset. But here the offset is uniformly sampled from $\{0, 1, \ldots, 100\}$.

For the model trained with the "w/o Pos." configuration, it is also tested without positional encodings. For the models trained with the rest configurations, they are all tested with the "w/ Pos." way of encoding.

The results are shown in Figure 4. The models in the left figure are trained on prompts shorter than 44, and the models in the right figure are trained on prompts longer than 45. We make the following observations. The pre-trained transformer in general can generalize to prompts with unseen length, under the condition of using/removing the positional embedding properly. The "w/o Pos." curve in the left figure shows that even at positions larger than 44, the model can still produce predictions close to Bayes-optimal. Adding positional encoding hurts the generalization ability. From the "w/ Pos." curve in the left figure, we find that the model's performance drops significantly at positions larger than 44. The main cause of the failure of length generalization is due to the distribution shift in the positional embedding space. As given in the "w/ S-Pos." and "w/ F-Pos." curves in the left figure, if the model has seen the *positional encodings* for a certain position during training, then its performance at this position is significantly improved, even if the *corresponding prompt length* is never seen. The length generalization ability is not unrestrictively strong, and such generalization ability for smaller lengths is generally weaker compared to that for larger lengths. The

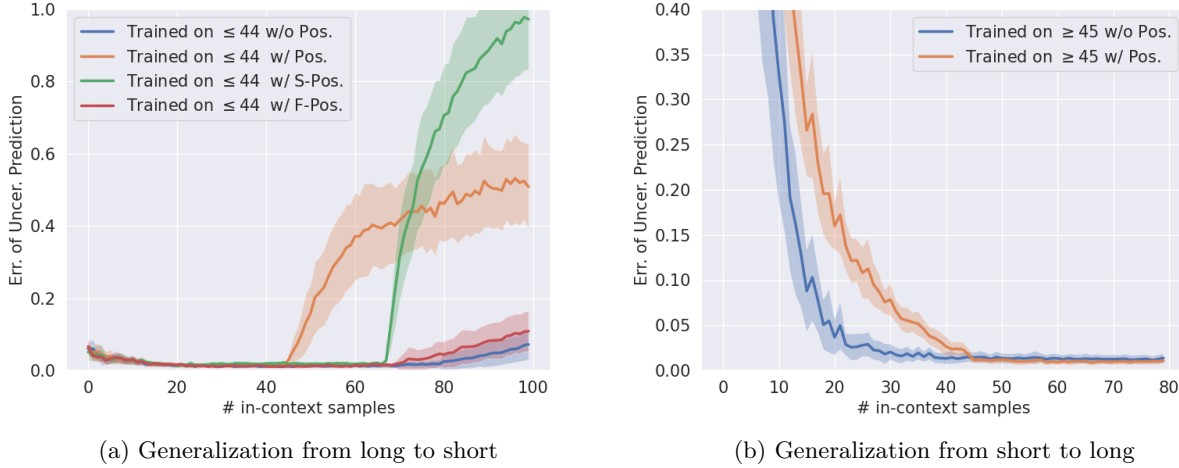

(a) Generalization from long to short

(b) Generalization from short to long

Figure 4: The effect of removing positional encoding on prompt length generalization. The *y*-axis records the average error of uncertainty prediction, which is the difference between the uncertainty predicted by the transformer and the Bayes-optimal estimator. (a) For models trained with prompt lengths $\leq 44$, the figure on the left shows that positional encoding has the worst generalization capacity with a larger length, and removing positional encoding could effectively enhance the length generalization power. (b) For models trained with prompt lengths $\geq 45$, removing positional encoding can help generalize to smaller lengths, although the generalization ability for smaller lengths is generally weaker compared to that for larger lengths.

right figure shows that even for the "w/o Pos." configuration, its performance still degrades when the prompt length is shorter than 20.

Wu et al. (2023)'s Theorem 5.3 points out that under the case of the single-layer linear-attention-only Transformer model on a linear regression task with Gaussian priors (without position encoding) if we train the model to only predict one single label after observing $T$ context exemplars, the optimally trained model under $T = T_1$ also performs well at the case $T = T_2$ (compared to the Bayes-optimal predictor for $T = T_2$) if $T_1$ and $T_2$ are close. In particular, such a property holds because the single-layer linear-attention structure constructs a "statistical shortcut" to achieve a near-optimal solution to the linear regression problem (of which the Bayes optimal predictor is an optimally tuned ridge regression). The ridge regression's regularizing parameter is related to the sequence length and the signal-to-noise ratio (SNR). If SNR is fixed and $T_1$ and $T_2$ are close, the Bayes-optimal predictors for $T_1$ and $T_2$ are close, implying that the "statistical shortcut" learned at $T_1$ still works in the case of $T_2$. This result implies the possibility of context length generalization by a simplified Transformer model due to shared structures in the attention matrices. The model of Wu et al. (2023) is also a simplified Transformer that skipped the position encoding part. Another point is that our empirical results also show that as the training and test context lengths get more different, the generalization becomes worse, which is of the same spirit as Wu et al. (2023).

## 5 Conclusion and Limitations

In this paper, we study the in-context learning ability of the trained Transformer through the lens of uncertainty quantification. In particular, we train the Transformer for a bi-objective task of mean prediction and uncertainty prediction. We develop new results both theoretically and numerically. The takeaway messages are: First, the Transformer can perform in-context uncertainty quantification. Second, the trained Transformer is only guaranteed to achieve a near-optimal in-distribution risk against the Bayes-optimal predictor. This does not imply that the Transformer behaves as the Bayes-optimal predictor either in-distribution or out-of-distribution. Third, the major concern of restricting the window size has been computational due to the quadratic growth of the computational consumption with respect to the context length, while our theory shows that limiting context length/window sizes can also benefit the generalization when the data size is limited.

Finally, the Transformer has the in-context ability for out-of-distribution tasks, but this in-context ability is contingent on a proper training method, such as sufficient task diversity, meta-training for covariate shift, and effective removal of the positional encoding. Two important future directions are as follows. First, we believe our method for deriving the generalization bound has implications for a scope much larger than uncertainty quantification and can be used to improve the existing bounds for various tasks using Transformers. Second, all the numerical experiments in the paper are conducted for the linear functions $f_i$'s. We believe the same results still hold for nonlinear functions as well; and such results can further consolidate the in-context ability for uncertainty quantification of the Transformer.

Our work also has limitations both theoretically and numerically. First, although Theorem 3.2 gives a tighter generalization bound when the context window $S$ is limited, it might not adapt to those LLMs that are of very large window sizes nowadays. However, it might still be valuable for those situations where smaller Transformers suffice. For example, for online decision-making problems where the action space and the sequence length are much smaller than the natural language processing problems, smaller models such as GPT-1 structure have been applied to construct Decision Transformer (Chen et al., 2021). Second, our work criticizing the Bayesian explanation of ICL ability itself does not give a positive explanation of the ICL dynamics; how ICL works remains unclear for future study. Finally, our discussions are limited to the regression tasks, while whether and how Transformers perform ICL in other tasks is beyond this work's scope.

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

## A Related Works

**Theoretical Understanding of In-Context Learning.** There are two streams of research in the theoretical understanding of ICL: the first tries to give sharp approximation error bounds on different tasks, while the second focuses on how the trained Transformer approaches the potential optimum. For the approximation error, following the pioneering empirical investigations on simple function classes (Garg et al., 2022), Von Oswald et al. (2023); Akyürek et al. (2022) conjecture that the Transformer is doing ICL via gradient descent, and verify it both empirically and theoretically. Based on the mechanism of layer-wise gradient descent construction, Bai et al. (2024) show that Transformers are able to behave (approximately) as well as some well-known algorithms on some statistical problems. Some following works generalize the layer-wise gradient descent construction to other settings such as decision-making (Lin et al., 2023a) and linear regression under representations (Guo et al., 2023). Apart from the layer-wise gradient descent, some other works consider the one-step gradient descent reached by a single-layer linear-activated Transformer (Zhang et al., 2023a; Wu et al., 2023) and curve the excessive population risk of the optimal model compared to oracle or the Bayes-optimal predictor. Ahn et al. (2024) give a set of global optima for some specific one-layer or two-layer attention-only models with linear or ReLU activation. Aside from characterizing where the Transformer *can* reach, another group of works is making efforts towards understanding where the Transformer *will* reach. One typical way is to study the simplified attention-only Transformers. Zhang et al. (2023a) start the analysis of the training dynamics of the gradient flow over the population risk on the linear regression task and show that a single-layer linear-attention-only model converges to some specific sets with suitable initialization. Wu et al. (2023) keep the same spirit and give a sample complexity bound based on a certain gradient descent scheme. For general Transformer models, technical tools from the statistical learning theory are applied. As for the task of predicting the next token in natural language tasks, Xie et al. (2021) provide a viewpoint from the Hidden Markov Model (HMM) and prove the asymptotic consistency under the regularity condition. Bai et al. (2024); Lin et al. (2023a) use chaining arguments with covering numbers for generalization, where Bai et al. (2024) consider the training under fixed length and Lin et al. (2023a) consider the problem of sequential decision-making. Li et al. (2023b) adopt algorithm stability arguments obtaining a bound of $\tilde{O}(1/\sqrt{nT})$. As is discussed in the main text (see discussions after Theorem 3.2), their analysis will result in a suboptimal $\tilde{\mathcal{O}}(S/\sqrt{nT})$ for the case $S < T$. Zhang et al. (2023b) adopt a similar concentration inequality for Markov chains to get a bound of $\tilde{\mathcal{O}}(\sqrt{\tau_{\mathrm{mix}}/(nT)})$. Since they do not consider the limit of context window $S$, their derivation ends up with $\tau_{\mathrm{mix}} \geq T$, which is suboptimal compared to our case. In short, our paper is the first theoretical analysis on the limit of context window $S$ and gets a tighter generalization bound than previous works on the generalization bound when $S < T$.

**Bayesian Behavior of In-context Learning.** Due to the complex structure of transformers, showing the theoretical properties of ICL without proper assumptions are challenging. There has been growing interest in

developing experiments to test various properties of ICL, leading to new observations and insights. Some of the earliest works that show transformers behave like Bayesian estimator can be found in Akyürek et al. (2022); Garg et al. (2022), and this argument is supported in follow-up works including Li et al. (2023b); Wu et al. (2023); Bai et al. (2024). However, there is also increasing empirical evidence demonstrating transformers' non-Bayesian behavior. Singh et al. (2024) design flipped experiment and show transformers' Bayesian behavior could be transient. Raventós et al. (2024); Panwar et al. (2023) demonstrate that the Bayesian behavior of transformers is dependent on the task diversity in the pretraining dataset, and transformers could deviate from the Bayesian predictor if number of different training tasks is large. Falck et al. (2024) design experiments based on the martingale property, a necessary condition of Bayesian behavior, and provide evidence that transformers exhibit non-Bayesian behavior from a statistical perspective.

**Transformers for Uncertainty Quantification.**   Uncertainty quantification has seen significant development within the general machine learning and deep learning domains (Abdar et al. (2021); Gawlikowski et al. (2023)), generating considerable interests within communities working on transformer-based large language models (LLMs). See Kuhn et al. (2023); Manakul et al. (2023); Lin et al. (2023b) for uncertainty quantification using black-box LLMs, and Slobodkin et al. (2023); Chen et al. (2024); Ahdritz et al. (2024) for that of white-box LLMs. Most of these works focus on natural language processing tasks that have less statistical properties. Indeed, uncertainty quantification has traditionally been developed from a more statistical and probabilistic perspective (Smith (2013); Sullivan (2015)). By adopting transformer models to study more statistics-related problems, our work aims to bridge and contribute to both fields.

We thank the reviewers for pointing out some literature for discussion. Although the problem setup seems to be similar to domain generalization/adaptation or credal set learning theory (Caprio et al., 2024), the main purpose of our work is different. The learner may face different types of distribution shifts between the training data and the test data, and those works in the previously mentioned field are trying to curve the learned results in those settings. However, our work aims to show that there is a gap between *Bayes optimal* and *Bayesian*. When infinite training data is available and facing ID test data, the Bayesian learner can behave as Bayes optimal. But the Bayes optimal may just be a result of ERM training as is shown in our work. We test the Transformers on OOD data to see if they coincide with Bayesian predictions. The answer is suspicious especially when the OOD issue becomes severe. Another line of research focus on quantifying the uncertainty, especially factorizing the aleatoric uncertainty (AU) and the epistemic uncertainty (EU) (Sale et al., 2023; Caprio et al., 2023; Wimmer et al., 2023). In this paper, our work differs from the aim of the above-mentioned uncertainty quantification literature. We do not manage each term accordingly (the AU and the EU). We are designing simple yet illustrative uncertainty quantification examples to test the ICL ability of Transformers rather than proving Transformers should be (one of) the most appropriate approach(es) to do uncertainty quantification.

## B   Transformer Model

Following Radford et al. (2019), we consider a decoder-only $L$-layer Transformer model that processes the input sequence $H_t$ by applying multi-head attention (MHA) of $M$ heads and multi-layer perceptron (MLP) layer-wise. Without loss of generality, we assume $x_t \in \mathbb{R}^d$ for some $d \geq 2$. We concatenate each $y_t$ with $d-1$ zeros so that it matches the format of each $x_t$, while we still denote the concatenated vector by $y_t$ with a slight abuse of notations. We denote $H_t$ by a matrix in $\mathbb{R}^{d \times (2t-1)}$ for $t = 1, \ldots, T$, where $H_t = [x_1, y_1, \cdots, x_t]$. We may also refer to $x_t$ by $h_{2t-1}$ and $y_t$ by $h_{2t}$. In practice, the attention mechanism has a maximum dependence length, and therefore the Transformer model can only produce an output based on the most recent tokens up to a context window size $S$. Hence we assume that at each time step $t$, the Transformer model has a maximum capacity of making predictions based on $x_t$ and previous $S$ pairs of $(x_s, y_s)$ observations for $s = t - S, \ldots, t - 1$. In other words, the Transformer has a maximum capacity of processing $2S + 1$ tokens, and it is making predictions $\texttt{TF}_\theta(H_t) = \texttt{TF}_\theta(H_t^S)$ based on the *truncated history* $H_t^S$, where $H_t^S := (x_{\max\{1, t-S\}}, y_{\max\{1, t-S\}}, \ldots, x_t)$. In the following, we formally describe the architecture of the Transformer used in this paper.

**Definition B.1** (Multi-Head Attention). A multi-head attention layer with $M$ heads and activation function $\texttt{act}(\cdot)$ can be defined as a function $\texttt{MHA}_W(\cdot)$ for any sequence $Z_t \in \mathbb{R}^{d \times (2t-1)}$ and $t = 1, \ldots, S+1$,

$$\texttt{MHA}_W(Z_t) = Z_t + \sum_{m=1}^{M} (W_V^m Z_t)\texttt{act}\big((W_K^m Z_t)^\top (W_Q^m Z_t)\big),$$

where $W = \{(W_Q^m, W_K^m, W_V^m)\}_{m=1}^{M}$ denotes all the parameters, $W_Q^m, W_K^m \in \mathbb{R}^{d_m \times d}$, $W_V^m \in \mathbb{R}^{d \times d}$ for each $m = 1, \ldots, M$, and $\texttt{act} \colon \mathbb{R}^{(2t-1) \times (2t-1)} \to \mathbb{R}^{(2t-1) \times (2t-1)}$ is the activation function.

Here we merge the residual connection into the multi-head layer and skip the layer normalization to ease the notations and simplify the analysis. The activation function is usually set to be columns-wise softmax in practice: for each vector $z \in \mathbb{R}^{2t-1}$,

$$\texttt{softmax}(z) := \left( \frac{\exp(z_1)}{\sum_{i=1}^{2t-1} \exp(z_i)}, \ldots, \frac{\exp(z_{2t-1})}{\sum_{i=1}^{2t-1} \exp(z_i)} \right)^\top.$$

Some theoretical results also consider alternative choices for $\texttt{act}$. For example, Akyürek et al. (2022); Ahn et al. (2024); Zhang et al. (2023a) consider the linear activation (that is, to entry-wise divide by the sequence length $2t-1$). Bai et al. (2024); Guo et al. (2023) also examine the $\texttt{ReLU}$ activation (that is, to entry-wise apply a $\texttt{ReLU}$ function $\texttt{ReLU}(z) = \max\{0, z\}$ and later divide by the sequence length $2t-1$).

**Definition B.2** (Multi-Layer Perceptron). A multi-layer perceptron layer with hidden dimension $d_h$ can be defined as a (token-wise) function $\texttt{MLP}_A(\cdot)$ for any sequence $Z_t \in \mathbb{R}^{d \times (2t-1)}$ and $t = 1, \ldots, S+1$,

$$\texttt{MLP}_A(Z_t) = Z_t + A_2 \texttt{ReLU}(A_1 Z_t),$$

where $A = (A_1, A_2)$ denotes all the parameters, $A_1 \in \mathbb{R}^{d_h \times d}$, $A_2 \in \mathbb{R}^{d \times d_h}$, and $\texttt{ReLU}$ is the entry-wise $\texttt{ReLU}$ function.

We merge the residual connection into the multi-layer perceptron layer and omit the layer normalization to simplify the theoretical development.

**Definition B.3** (Transformer). A Transformer model with $L$ layers can be defined as a function $\texttt{TF}_\theta(\cdot)$ for any sequence $Z_t \in \mathbb{R}^{d \times (2t-1)}$ and $t = 1, \ldots, S+1$. For the $l$-th layer, the model receives $Z_t^{(l-1)}$ as the input and processes it by an $\texttt{MHA}$ block and an $\texttt{MLP}$ block, such that

$$Z_t^{(l)} = \texttt{MLP}_{A^{(l)}}(\texttt{MHA}_{W^{(l)}}(Z_t^{l-1})), \quad \forall l = 1, \ldots, L,$$

where $Z_t^{(0)} = Z_t$. After the $L$-th layer, the model linearly maps the $Z_t^{(L)} \in \mathbb{R}^{d \times (2t-1)}$ onto $\mathbb{R}^{2 \times (2t-1)}$ via a matrix $P \in \mathbb{R}^{2 \times d}$, and we process the second dimension by a $\texttt{softplus}$ function to get the final prediction as

$$\hat{y}_\theta(Z_t) = (P Z_t^{(L)})_{1,2t-1},$$

and

$$\hat{\sigma}_\theta(Z_t) = \texttt{softplus}\big((P Z_t^{(L)})_{2,2t-1}\big).$$

Here $\theta = (\{(W^{(l)}, A^{(l)})\}_{l=1}^{L}, P)$ encapsulates all the parameters and the function $\texttt{softplus}(z) = \log(1 + \exp(z))$ is introduced to avoid negative output. The output is summarized as $\texttt{TF}_\theta(Z_t) := (\hat{y}_\theta(Z_t), \hat{\sigma}_\theta(Z_t))$.

*Remark* B.4. To enable parallel training, the decoder-only Transformer receives a full sequence in the training phase. The model has a masking component that prevents the model from seeing into the "future". However, such masking is unnecessary in our setting as the Transformer model receives exactly what it should "see" at each time $t$, and the full dynamics are identical to those in the masked setting.

**Miscellaneous notations.** Denote the set $\{1, \ldots, K\}$ by $[K]$. Denote the consecutive sequence $\{i, i + 1, \ldots, j\}$ by $i : j$. Denote the matrix $A$'s entry at the $i$-th row and the $j$-th column by $A_{i,j}$. Denote the vector $x$'s $i$-th element by $(x)_i$. Define the $d$-dimensional vector $x$'s $p$-norm as $(\sum_{i=1}^{d}(x)_i^p)^{1/p}$ for $p \in [1, \infty]$, where $\|x\|_\infty = \max_{1 \le i \le d}(x)_i$. Define the $m \times n$-sized matrix $A$'s $(p, q)$-norm as $(\sum_{j=1}^{n} \|A_{:,j}\|_p^q)^{1/q}$. Denote the $d$-dimensional diagonal matrix by $\mathrm{diag}\{\lambda_1, \ldots, \lambda_d\}$. Denote the $d$-dimensional identity matrix by $I_d$. Denote the total variation distance between two probability distributions $P$ and $Q$ by $\mathrm{TV}(P, Q)$. Denote the Kullback-Leibler divergence between two probability distributions such that $P \ll Q$ by $D_{\mathrm{kl}}(P\|Q)$. Denote the product measure of $P$ and $Q$ by $P \times Q$ or $P \otimes Q$. Denote the Cartesian product of two spaces $\mathcal{X}$ and $\mathcal{Y}$ by $\mathcal{X} \times \mathcal{Y}$. Denote the tensor-product $\sigma$-algebra of two $\sigma$-algebras $\Sigma_1$ and $\Sigma_2$ by $\Sigma_1 \otimes \Sigma_2$. Denote the limiting behavior of being upper (lower, both upper and lower, respectively) bounded by up to some constant(s) by $\mathcal{O}$ ($\Omega$, $\Theta$, respectively). Denote $\tilde{\mathcal{O}}$ to be the $\mathcal{O}$ but omitting some poly-logarithmic terms.

## B.1 Assumptions

Based on this setup of the Transformer model, we introduce the following bounded assumptions used for Theorem 3.2. Such assumptions are common in the analyses of the Transformer model (Bai et al., 2024; Zhang et al., 2023b) by either assuming an extra clipping operator or explicit upper bounds.

**Assumption B.5.** Assume $\Theta = \mathcal{B}(0, B_{\mathrm{TF}})$, where the norm is defined as

$$\|\theta\| := \max\{\|W^{(l)}\|, \|A^{(l)}\|, \|P\| \; : \; l = 1, \ldots, L\}.$$

The corresponding norms are defined as

$$\|W\| := \max\{\|W_V^m\|_{2,2}, \|W_K^m\|_{2,2}, \|W_Q^m\|_{2,2} \; : \; m = 1, \ldots, M\},$$

$$\|A\| := \max\{\|A_1\|_{2,2}, \|A_2\|_{2,2}\}, \quad \|P\| := \|P^\top\|_{2,\infty},$$

where we omit some superscripts/subscripts of the layer number ($l$) for simplicity.

**Assumption B.6.** Assume $\|H_t\|_{2,\infty}$ is bounded by $B_H$ almost surely. Such a regularization is equivalent to assuming $\|x_t\|_2 \le B_H$ and $|y_t| \le B_H$ almost surely.

## B.2 Approximation Error

In Section 3, we provide the generation bound in Theorem 3.2. Now we give an analysis for the approximation error. We define the Bayes-optimal risk obtained by the Bayes-optimal predictor in Proposition 3.1: for each $t = 1, \ldots, T$,

$$R_t^* := \mathbb{E}\left[\ell\left(\left(y_t^*(H_t), \sigma_t^*(H_t)\right), y_t\right)\right]. \tag{4}$$

However, Transformers only have access to the truncated history $H_t^S$, which prevents them from reaching $R_t^*$. By using Proposition 3.1 for the $H_t^S$, we denote the *truncated* Bayes optimum for each $t$:

$$y_t^{S*}(H_t^S) := \mathbb{E}[y_t|H_t^S],$$

and

$$\left(\sigma_t^{S*}(H_t^S)\right)^2 := \mathbb{E}[(f(x_t) - y_t^{S*}(H_t^S))^2|H_t^S] + \mathbb{E}[\sigma^2|H_t^S].$$

We denote the truncated Bayes-optimal risk as

$$R_t^{S*} := \mathbb{E}\left[\ell\left(\left(y_t^{S*}(H_t^S), \sigma_t^{S*}(H_t^S)\right), y_t\right)\right]. \tag{5}$$

It is straightforward to check that

$$R_t^{S*} = R_t^*, \quad \text{for any } t \le S. \tag{6}$$

However, the equality is generally not true for $t > S$. We give an example to illustrate the gap.

**Example B.7.** *Consider the case where one has oracle access to the noise level $\sigma$. Note that the oracle knowledge only reduces the risk $R_t^{S*}$, since we use information that is not a measurable function of $H_t^S$. The problem is reduced to a regression problem.*

*Suppose the function $f$ is linear and its weight vector has a prior distribution of $\mathcal{N}(0, \sigma^2 I_d)$, and the noise $\epsilon \sim \mathcal{N}(0, 1)$. Suppose $x_t \sim \mathcal{N}(0, I_d)$. Then the optimal estimator is an optimally tuned Ridge regression.*

*Tsigler & Bartlett (2023) show that with high probability, the optimal Ridge regression estimator has an average risk of $\frac{1}{2} + \Theta(1/S)$, where the term $\frac{1}{2}$ is due to $\mathbb{E}[(y - f(x))^2]/(2\sigma^2)$. But as the length $t$ approaches infinity, the average risk of the optimal Ridge regression over the full sequence $H_t$ will converge to $\frac{1}{2}$ with high probability, meaning that the estimated $\hat{f}$ will converge to true $f$ for every sequence. Hence one can always construct an uncertainty estimation by averaging all the residuals, and such an estimation $\hat{\sigma}$ will converge to the true $\sigma$. Thus, we have $R_t^*$ approaching $\frac{1}{2}$ as $t$ grows to infinity, leading to the conclusion that*

$$R_t^{S*} - R_t^* \geq \Omega(1/S),$$

*for sufficiently large $t$.*

Example B.7, together with Theorem 3.2, shows the approximation-estimation tradeoff in selecting the context window $S$ of Transformer models. Previous works (Wu et al., 2023; Bai et al., 2024; Zhang et al., 2023b; Guo et al., 2023) consider the case where $t \leq S$, and establish the upper bounds for the approximation error. In other words, these existing results are all made with respect to the gap between $R_t^{S*}$ and $R(\text{TF}_{\theta^*})$. To our knowledge, we are the first work to point out the extra term of approximation error due to truncation.

## C  More Numerical Results and Discussions

### C.1  In-distribution performance

In Figure 1, we provide a comparison of the in-distribution performance of the trained Transformer v.s. the Bayes-optimal predictor. A subtle point is that for the uncertainty prediction, we only plot the average predicted uncertainty, this does not fully imply that the Transformer gives a similar prediction as the Bayes-optimal predictor. To this end, Figure 5 plots the difference between each of the models and the Bayes-optimal predictor in terms of uncertainty estimation.

### C.2  Out-of-distribution perfomance

In Figure 2, we plot the Bayes-optimal predictor under three OOD settings, and we note that though the Bayes-optimal predictor uses a wrong prior, it has the ability to work as an algorithm to correct the prediction with the in-context samples. Now in Figure 6 (a), we compare the Bayes-optimal predictor that uses the wrong prior with the Bayes-optimal predictor that uses the correct prior (which replaces the in-distribution prior with the correct OOD prior of $\sigma^2$). The figure is based on the large OOD setting. We observe that the Bayes-optimal predictors with the ID prior or the OOD prior both converge to the true uncertainty level. For Figure 6 (b), we plot the performances under the same large OOD setting. As a reference line, we copy and paste the Bayes-optimal predictor's curve in Figure 1 (b) here. We note this reference line is computed based on the in-distribution (ID) data and is not comparable at all to the predicted uncertainty level on the OOD data. Yet, we note that when the number of tasks is small when training the Transformer (say $N = 4096$), it tends to make predictions on the OOD data by treating the OOD data just as ID data, and this means the trained Transformer is doing in-weight learning and has no in-context learning ability. As the number of tasks increases, the Transformer gradually gains the in-context ability and moves towards the Bayes-optimal predictor on the OOD data.

### C.3  Training dynamics and task shift OOD performance

Now we zoom into the training dynamics to further investigate the OOD performance under task shift. In the following example, we derive a theoretical result based on Theorem 4.1 in Zhang et al. (2023a). Specifically, $R$ and $R'$ (following the notations therein) denote the in-distribution and out-of-distribution expected risk.

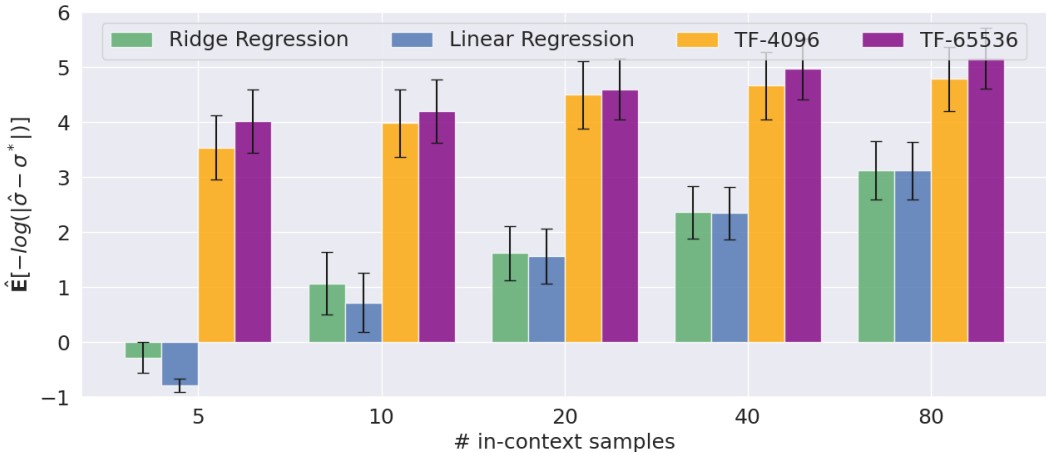

Figure 5: In-distribution performance of the uncertainty prediction against the Bayes-optimal predictor. The $y$-axis gives an estimate of $\mathbb{E}\left[-\log|\hat{\sigma}(H_t) - \sigma^*(H_t)|\right]$ where the expectation is taken with respect to $H_t$. Here $\hat{\sigma}(H_t)$ is the uncertainty estimate produced by an algorithm (ridge regression, linear regression, or transformer), and $\sigma^*(H_t)$ is the Bayes-optimal predictor given in Proposition 3.1 and calculated by Section G.3. The figure shows that the Transformer and the Bayes-optimal predictor produce similar uncertainty predictions. In addition, the Transformer trained on a larger pool of tasks (larger $N$) produces a better approximation of the Bayes-optimal predictor.

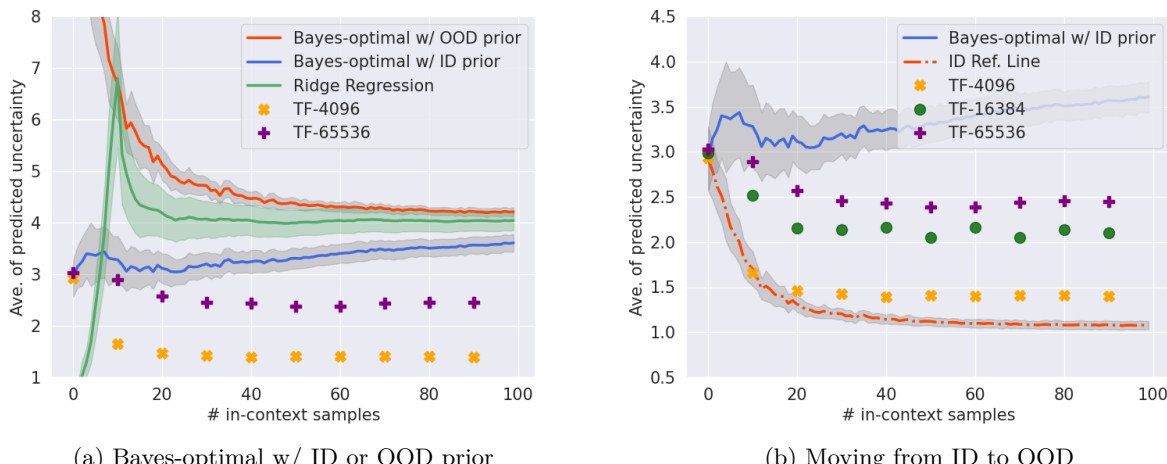

(a) Bayes-optimal w/ ID or OOD prior          (b) Moving from ID to OOD

Figure 6: Performance under L-OOD setting. For both (a) and (b), the $y$-axis gives the average of the predicted uncertainty over all the test samples (average of $\hat{\sigma}(H_t)$ or $\sigma^*(H_t)$ on test samples), and ideally the curves should converge to the true uncertainty level of 4 as the number of in-context samples increases. In (a), we compare the Bayes-optimal predictor that uses the wrong prior with the Bayes-optimal predictor that uses the correct prior (which replaces the in-distribution prior with the correct OOD prior of $\sigma^2$). Both work well in that the curves converge to the true mean uncertainty level of around 4. The Transformers deviate from both Bayes-optimal predictors due to the large OOD intensity. In (b), we observe that as the training task diversity increases. The transformer gradually moves from the ID reference line to the Bayes-optimal predictor.

The result says that while the in-distribution risk continues decreasing over time, the out-of-distribution risk may keep increasing or may first decrease and then increase. Importantly, the out-of-distribution risk may depend on the initial point of the training procedure.

Reaching the Bayes optimum requires prior knowledge of the underlying distribution. We provide a simple example of where the Transformer stores its prior knowledge and how it hurts the OOD performance even under a mild distribution shift.

**Example C.1** (A corollary that can be derived based on Theorem 4.1 in Zhang et al. (2023a)). *Consider a one-layer attention-only Transformer model with linear activation and one attention head on the linear regression task. We now concatenate each of the inputs to be $[x_t^\top, y_t]^\top \in \mathbb{R}^{d+1}$. Suppose we focus on the linear regression task on the $(T+1)$-th sample after observing $T$ context exemplars, where each $w^{(i)} \sim \mathcal{N}(0, I_d)$, each $x_t^{(i)} \sim \mathcal{N}(0, I_d)$, each $\epsilon_t^{(i)} \sim \mathcal{N}(0,1)$, and $y_t^{(i)} = w^{(i)\top} y_t^{(i)} + \sigma_0 \cdot \epsilon_t^{(i)}$. If we adopt the same training setup as Zhang et al. (2023a) (with details referred to therein), then for any $|\sigma_0' - \sigma_0| \geq \Delta$ for some $\Delta > 0$, if we train on the distribution w.r.t. $\sigma_0$ but test on the distribution w.r.t. $\sigma_0'$ (denoted by $R'$), then for $C = d/(16(2+\sigma_0))$ and any sequence $0 < \delta_1 < \delta_2 < \cdots < C\Delta$, there exists a non-decreasing sequence $0 \leq \tau(\delta_1) \leq \tau(\delta_2) \leq \ldots$, such that*

$$R'(\tau(\delta_i)) - R'_{\theta^{*\prime}} \geq \delta_i, \quad \text{for each } i = 1, 2, \ldots,$$

*while the parameter $(W_K^\top W_Q)_{1:d,1:d}(W_V)_{d+1,d+1}$ converges to $1/(1+(2+\sigma_0)/T) \cdot I_d$ (which is the corresponding part of some $\theta^*$). Here $\theta^*$ and $\theta^{*\prime}$ minimize the population risk $R$ and $R'$, accordingly.*

We design an experiment to show that as training proceeds, the model's OOD performance is improved abruptly in the starting phase, but then degrades steadily after too many steps of training. We introduce the experiment settings below. A visualization of the setup is given in Figure 7.

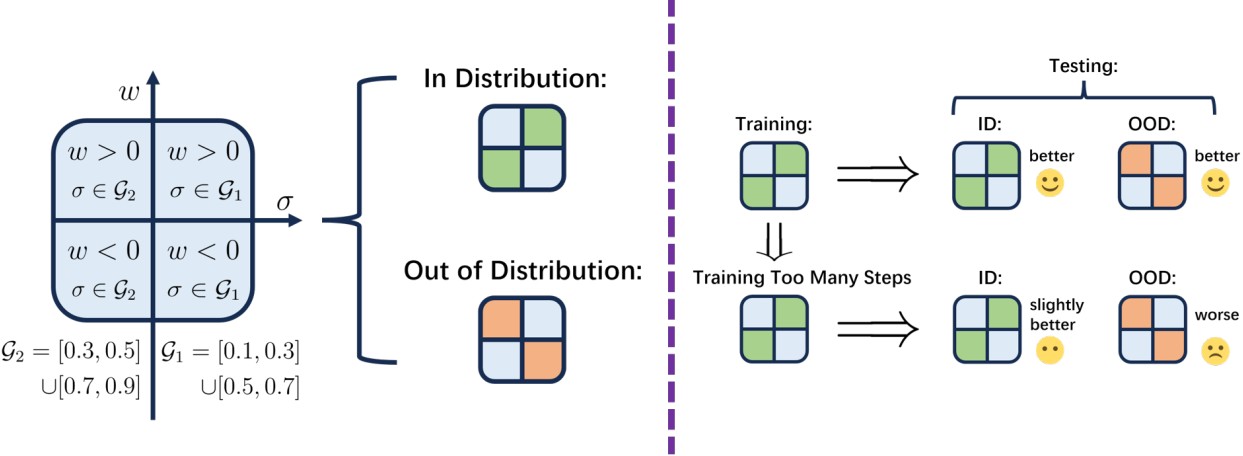

Figure 7: The settings of the OOD experiment. (Left) The in-distribution (ID) tasks are sampled from regions denoted by the green blocks, and the OOD tasks are sampled from the red blocks. (Right) In the starting phase, training improves both the ID and OOD performance. But if training for too many steps, the ID performance is only marginally improved, while the OOD performance steadily degrades.

Each linear task in our uncertainty quantification setting is characterized by parameters $(w, \sigma)$. We define two regions for $w$, denoted by $\mathcal{W}_1$ and $\mathcal{W}_2$. And two regions for $\sigma$, denoted by $\mathcal{G}_1$ and $\mathcal{G}_2$. When $w$ is sampled from $\mathcal{W}_1$, $w$ follows the following distribution

$$w = |\beta|, \quad \beta \sim \mathcal{N}(0, I_8).$$

When $w$ is sampled from $\mathcal{W}_2$, $w$ follows the following distribution

$$w = -|\beta|, \quad \beta \sim \mathcal{N}(0, I_8).$$

For $\sigma$, define $\mathcal{G}_1 = [0.1, 0.3] \cup [0.5, 0.7]$ and $\mathcal{G}_2 = [0.3, 0.5] \cup [0.7, 0.9]$. Define $\mathcal{G}_1$ and $\mathcal{G}_2$ to be the "complementary" group of each other. We sample $\sigma$ independently from $w$, and we always sample $\sigma$ uniformly from either group $\mathcal{G}_1$ or $\mathcal{G}_2$. As marked in Figure 7, the "ID" tasks sample its parameters from $(w, \sigma) \in \mathcal{W}_1 \otimes \mathcal{G}_1 \bigcup \mathcal{W}_2 \otimes \mathcal{G}_2$ and the "OOD" tasks sample its parameters from $(w, \sigma) \in \mathcal{W}_1 \otimes \mathcal{G}_2 \bigcup \mathcal{W}_2 \otimes \mathcal{G}_1$. The training is on "ID" tasks, and

the trained model is tested on both "ID" tasks and "OOD" tasks. The metric we evaluate in this experiment is the "prediction accuracy" of uncertainty. The accuracy denotes the probability that the model predicts the $\sigma$ into its "*right*" group. (for a prompt generated from $(w, \sigma)$ with $\sigma \in \mathcal{G}$, we say that the model makes a "right" prediction if the predicted $\sigma$ falls into $\mathcal{G}$).

Figure 8 presents the experiment result. The prediction accuracy on the ID dataset peaks after 20k steps of training. At the same time, the prediction accuracy on the OOD dataset also increases to 80%. After that, the ID performance remains unchanged, but the OOD accuracy keeps dropping.

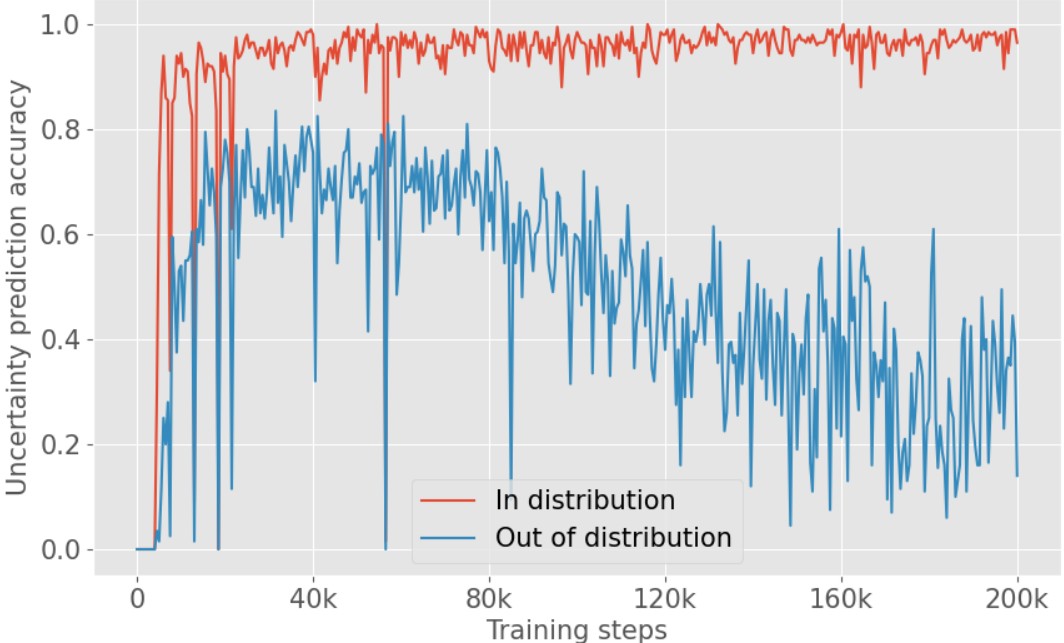

Figure 8: The accuracy denotes the probability that the model predicts the $\sigma$ into the "right" group. For example, if the sampled tasks take $\sigma$ from group $\mathcal{G}_1$, then accuracy denotes the probability that the model predicts $\sigma$ into $\mathcal{G}_1$. The data is collected for the 100-th token in order to eliminate the epistemic uncertainty due to insufficient in-context samples. The x-axis denotes the training steps. This figure shows that when training too many steps ($> 40k$ in this case), the generalization ability of the model steadily declines.

In order to verify that the degradation of OOD performance is due to the increasing confidence in the prior information of the training data, we check for the OOD distribution whether the model has predicted $\sigma$ into the complementary group. The result is presented in Figure 9, which verifies that after training too many steps, the model tends to predict $\sigma$ following the training prior. A more concrete way to explain it: consider an OOD sampled task $(w, \sigma)$ where $w > 0$. According to the sampling rule of OOD tasks, it must have $\sigma \in \mathcal{G}_2$. If the model has the OOD ability, it should predict $\sigma \in \mathcal{G}_2$. But if it has too much confidence in its training prior, it will predict $\sigma$ into $\mathcal{G}_1$, the complementary group of $\mathcal{G}_2$. Figure 9 shows that for the misclassified OOD tasks, the model has predicted them into complementary groups.

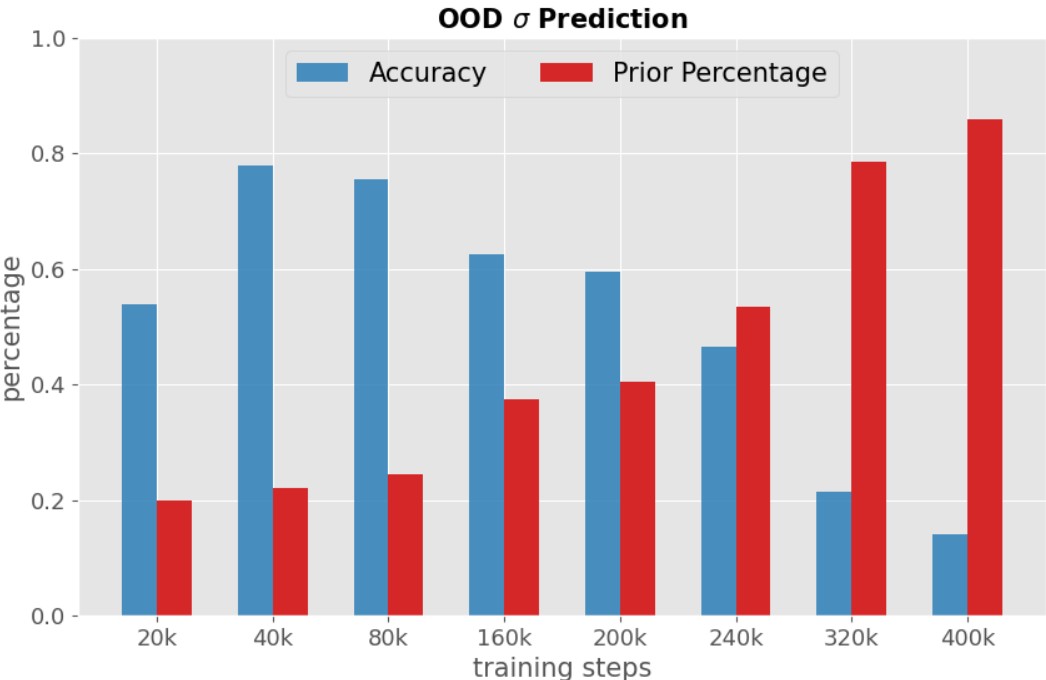

Figure 9: This figure validates that the decline of OOD ability is due to increasing confidence in the training prior. The blue bars correspond to the OOD accuracy, and the red bars give the probability that the model predicts uncertainty $\sigma$ into the complementary group (i.e. the training distribution of $\sigma$). As the training proceeds, most of the misclassified $\sigma$ are predicted following the training prior.

# D Proofs of the Results in the Main Paper

## D.1 Proof of Proposition 3.1

*Proof.* Recall that the population risk is

$$L(\hat{y}, \hat{\sigma}) := \mathbb{E}_{f, x_{[t]}, \epsilon_{[t]}, \sigma}\left[\log \hat{\sigma}(H_t) + \frac{(y_t - \hat{y}(H_t))^2}{2\hat{\sigma}^2(H_t)}\right].$$

We first prove that for any $\hat{\sigma}(H_t)$, the choice of $\hat{y}_t = y^* = \mathbb{E}[y_t|H_t]$ minimizes the population risk. With any fixed $\sigma_0 > 0$, when $\hat{\sigma}_t = \sigma_0$, then minimizing the population risk reduces to minimizing $\mathbb{E}[(y_t - \hat{y}(H_t))^2]$. Using Fubini's Theorem and the fact that the conditional distribution exists, we have

$$\mathbb{E}_{f, x_{[t]}, \epsilon_{[t]}, \sigma}\left[(y_t - \hat{y}(H_t))^2\right] = \mathbb{E}_{H_t}\left[\mathbb{E}\left[(y_t - \hat{y}(H_t))^2\big|H_t\right]\right]$$

$$= \mathbb{E}_{H_t}\left[\mathbb{E}\left[(f(x_t) - \hat{y}(H_t))^2\big|H_t\right]\right] + \mathbb{E}_{H_t}\left[\mathbb{E}\left[\sigma^2|H_t\right]\right], \quad (7)$$

where the last equality follows from the fact that $\epsilon_t$ is independent of $H_t$ and $\sigma$ and is of zero mean and unit variance. Since the second term on the right-hand-side of equation 7 does not depend on $\hat{y}$, we only need to focus on the first term. For each realization of $H_t$, the prediction $\hat{y}(H_t)$ is a single point; combining it with the fact that the squared loss is minimized with respect to one single point prediction if and only if that point is the expectation (in this case, the conditional expectation $\mathbb{E}[f(x_t)|H_t]$), we prove that for any $\sigma_0$, the population risk's minimizer

$$y_t^*(\sigma_0) = \mathbb{E}[f(x_t)|H_t] = \mathbb{E}[f(x_t) + \sigma \cdot \epsilon_t|H_t] = \mathbb{E}[y_t|H_t],$$

where the second equality follows again from the fact that $\epsilon_t$ is independent of $H_t$ and $\sigma$, and $\epsilon_t$ is of zero mean. Since this equality holds for an arbitrary $\sigma_0$, we can conclude that

$$y_t^* = \mathbb{E}[y_t | H_t].$$

Now we have confirmed the optimal choice of $y_t^*$ regardless of whatever $\hat{\sigma}$ is. We can thus find the optimal choice of $\hat{\sigma}$ by fixing $\hat{y} = y_t^*$ and minimizing the population risk. Similarly, we can change the integration order so that we only need to minimize $\mathbb{E}[\log \hat{\sigma}(H_t) + \frac{(y_t - \hat{y}(H_t))^2}{2\hat{\sigma}^2(H_t)} | H_t]$ for any realization of $H_t$. Calculations show that

$$\frac{\partial \mathbb{E}\left[\log \hat{\sigma}(H_t) + \frac{(y_t - \hat{y}(H_t))^2}{2\hat{\sigma}^2(H_t)} \mid | H_t\right]}{\partial \hat{\sigma}(H_t)} = \frac{\partial \left(\log \hat{\sigma}(H_t) + \frac{\mathbb{E}[(y_t - \hat{y}(H_t))^2 | H_t]}{2\hat{\sigma}^2(H_t)}\right)}{\partial \hat{\sigma}(H_t)}$$

$$= \frac{\hat{\sigma}^2(H_t) - \mathbb{E}[(y_t - \hat{y}(H_t))^2 | H_t]}{\hat{\sigma}^3(H_t)},$$

where the first equality follows from the fact that on observing $H_t$, $\hat{\sigma}(H_t)$ is a fixed value, and the second equality from the calculus. Thus, the risk is minimized if and only if $\hat{\sigma}(H_t) = \mathbb{E}[(y_t - \hat{y}(H_t))^2 | H_t]$. Substituting $\hat{y}$ for $y_t^*$, we have

$$\sigma_t^{*2}(H_t) = \mathbb{E}[(y_t - y_t^*(H_t))^2 | H_t] = \mathbb{E}[(f(x_t) - y_t^*(H_t))^2 | H_t] + \mathbb{E}[\sigma^2 | H_t],$$

where the last equality follows again from the fact that $\epsilon_t$ is independent of $H_t$ and is of zero mean and unit variance. $\square$

### D.2 Proof of Theorem 3.2

*Proof.* $\hat{\theta}^{\mathrm{ERM}}$ Before we start the detailed proof, we define another flattened sequence $(\tilde{x}_k, \tilde{y}_k)$ for $k = 1, \ldots, nT$, where for $k = iT + t$ we have

$$\left(\tilde{x}_{iT+t}, \tilde{y}_{iT+t}\right) := \left(x_t^{(i)}, y_t^{(i)}\right). \tag{8}$$

Here, we merge all the sequences $\{(x_t^{(i)}, y_t^{(i)})\}_{t=1}^T$ for $i = 1, \ldots, n$ into one sequence $(\tilde{x}_k, \tilde{y}_k)_{k=1}^{nT}$. Similarly, we can define a flattened truncated history $\tilde{H}_k^S$ as

$$\tilde{H}_{iT+t}^S := (x_{\max\{t-S,1\}}^{(i)}, y_{\max\{t-S,1\}}^{(i)}, \ldots, x_t^{(i)}, y_t^{(i)}. \tag{9}$$

Note that $\tilde{H}_{k,k=iT+t}^S = (H_t^{S(i)}, y_t^{(i)})$, since we have added the target label $y_t^{S(i)}$ into the flattened truncated history $\tilde{H}_k^S$ for notation simplicity. With a slight abuse of notations, we have

$$\ell_\theta(\tilde{H}_{k,k=iT+t}^S) := \ell(\mathrm{TF}_\theta(H_t^S), y_t) = \ell(\mathrm{TF}_\theta(H_t), y_t), \tag{10}$$

where the equality holds since we are making predictions based on at most $S$ pairs of $(x_t, y_t)$. We can similarly replace the $\ell$ function in the definition of empirical risk $r$ and population risk $R$, obtaining

$$r(\mathrm{TF}(\theta)) = \frac{1}{nT} \sum_{i=1}^n \sum_{t=1}^T \ell(\mathrm{TF}_\theta(H_t^S), y_t)$$

$$= \frac{1}{nT} \sum_{k=1}^{nT} \ell_\theta(\tilde{H}_k^S), \tag{11}$$

and

$$R(\mathrm{TF}(\theta)) = \frac{1}{T} \mathbb{E}_{H_t} \left[\sum_{t=1}^T \ell(\mathrm{TF}(\theta)(H_t), y_t)\right]$$

$$= \frac{1}{T} \mathbb{E}_{\tilde{H}_k^{S'}} \left[\sum_{t=1}^T \ell_\theta(\tilde{H}_{k,k=iT+t}^{S'})\right]$$

$$= \mathbb{E}_{\tilde{H}_k^{S'}} \left[\frac{1}{nT} \sum_{k=1}^{nT} \ell_\theta(\tilde{H}_k^{S'})\right], \tag{12}$$

where $\tilde{H}_t^{S\prime}$ is another flattened truncated history that is i.i.d. to $\tilde{H}_t^S$. For notation simplicity, we define

$$\tilde{\mathcal{H}}^S := (\tilde{H}_1^S, \ldots, \tilde{H}_{nT}^S). \tag{13}$$

Then we simplify the notations as

$$r_\theta(\tilde{\mathcal{H}}^S) := r(\mathtt{TF}(\theta)), \tag{14}$$

and

$$R_\theta := \mathbb{E}_{\tilde{\mathcal{H}}^{S\prime}}\big[r_\theta(\tilde{\mathcal{H}}^{S\prime})\big] = R(\mathtt{TF}(\theta)). \tag{15}$$

To control the difference between $R_\theta$ and $r_\theta(\tilde{\mathcal{H}}^S)$ for any $\theta$ (which could potentially depend on training data $\mathcal{D}$), we use PAC-Bayes arguments for simplicity.

All the following arguments are made with the conditional distribution on knowing each $f^{(i)}$ and $\sigma^{(i)}$, for each $i = 1, \ldots, n$. We omit the conditional dependencies in our notations only for simplicity.

By our definition of data generation, the flattened truncated history $\tilde{H}_k^S$ naturally forms up a Markov chain on the space $\otimes_{k=1}^{nT} \Omega_k$ (verified in Lemma E.13), since the newly generated $(x_t, y_t)$ are conditionally independent of all previous observations. Here $\Omega_{k,k=iT+t} := (\mathcal{X} \times \mathcal{Y})^{\otimes \min\{t,S\}}$.

Fix a $\theta$ that does not depend on the training data $\mathcal{D}$. We now bound the difference between $R_\theta$ and $r_\theta(\tilde{\mathcal{H}}^S)$ via concentration inequality for Markov chains. From Lemma F.2, we know that if the Markov chain's mixing time is small enough (which means it quickly converges to the stationary distribution), the concentration properties over the Markov chain would be good enough to enable the standard PAC-Bayes arguments. We also know from Lemma E.15 that the flattened truncated history has a mixing time no greater than $\min\{S, T\}$, since all the histories $S$ pairs before the current time would be truncated from the input, and the history $H_t^S$ restarts every time a sequence reaches length $T$. With these observations, we start our detailed derivation.

Since the function $\ell$ is almost surely bounded by $C_2$ as is shown in Lemma E.3, we have almost surely for any $\tilde{\mathcal{H}}^S$ and $\tilde{\mathcal{H}}^{S\prime}$,

$$r_\theta(\tilde{\mathcal{H}}^S) - r_\theta(\tilde{\mathcal{H}}^{S\prime}) \leq \sum_{k=1}^{nT} \frac{2C_2}{nT} \cdot \mathbb{1}\{\tilde{H}_k^S \neq \tilde{H}_k^{S\prime}\}. \tag{16}$$

We can use McDiarmid type's inequality for Markov chains (Lemma F.2, with the mixing time upper bound no greater than $\min\{S, T\}$ (specified in Lemma E.15), such that for any $\lambda \in \mathbb{R}$,

$$\mathbb{E}_{\mathcal{D}}\Big[\exp\big(\lambda(r_\theta(\tilde{\mathcal{H}}^S) - R_\theta(\tilde{\mathcal{H}}^S))\big)\Big] \leq \exp\Big(\frac{2\lambda^2 C_2^2 \min\{S,T\}}{nT}\Big). \tag{17}$$

Set $\pi$ to be the distribution over $\Theta$ defined in Lemma E.11. Since $\pi$ is chosen independently from $\mathcal{D}$, we can integrate equation 17 with respect to $\theta \sim \pi$ such that

$$\mathbb{E}_{\theta \sim \pi}\bigg[\mathbb{E}_{\mathcal{D}}\Big[\exp\big(\lambda(r_\theta(\tilde{\mathcal{H}}^S) - R_\theta(\tilde{\mathcal{H}}^S))\big)\Big]\bigg] \leq \exp\Big(\frac{2\lambda^2 C_2^2 \min\{S,T\}}{nT}\Big). \tag{18}$$

Using Fubini's Theorem, we can exchange the order of integration, such that

$$\mathbb{E}_{\mathcal{D}}\bigg[\mathbb{E}_{\theta \sim \pi}\Big[\exp\big(\lambda(r_\theta(\tilde{\mathcal{H}}^S) - R_\theta(\tilde{\mathcal{H}}^S))\big)\Big]\bigg] \leq \exp\Big(\frac{2\lambda^2 C_2^2 \min\{S,T\}}{nT}\Big). \tag{19}$$

By applying Donsker-Varadhan's formula (Lemma F.3), we derive from equation 19 that

$$\mathbb{E}_{\mathcal{D}}\bigg[\exp\Big(\sup_{\rho \in \mathcal{P}(\Theta)}\big\{\mathbb{E}_{\theta \sim \rho}\big[\lambda(r_\theta(\tilde{\mathcal{H}}^S) - R_\theta(\tilde{\mathcal{H}}^S)\big] - D_{\mathrm{kl}}(\rho \| \pi)\big\}\Big)\bigg] \leq \exp\Big(\frac{2\lambda^2 C_2^2 \min\{S,T\}}{nT}\Big).$$

Rearranging terms, we have

$$\mathbb{E}_{\mathcal{D}}\bigg[\exp\Big(\sup_{\rho \in \mathcal{P}(\Theta)}\big\{\mathbb{E}_{\theta \sim \rho}\big[\lambda(r_\theta(\tilde{\mathcal{H}}^S) - R_\theta(\tilde{\mathcal{H}}^S)\big] - D_{\mathrm{kl}}(\rho \| \pi)\big\} - \frac{2\lambda^2 C_2^2 \min\{S,T\}}{nT}\Big)\bigg] \leq 1. \tag{20}$$

Using Chernoff's bound (Lemma F.4) with probability $\delta/4$, we have with probability at least $1 - \frac{\delta}{4}$ w.r.t. $\mathcal{D}$,

$$\sup_{\rho \in \mathcal{P}(\Theta)} \left\{ \mathbb{E}_{\theta \sim \rho} \left[ \lambda(r_\theta(\tilde{\mathcal{H}}^S) - R_\theta(\tilde{\mathcal{H}}^S)) \right] - D_{\mathrm{kl}}(\rho\|\pi) \right\} - \frac{2\lambda^2 C_2^2 \min\{S,S\}}{nT} \le \log(4/\delta). \tag{21}$$

Since this bound equation 21 holds for *any* distribution $\rho$ over $\Theta$, we can set $\rho$ to be $\rho_{\hat{\theta}^{\mathrm{ERM}}}$ as defined in Lemma E.11, resulting in a high-probability bound

$$
\begin{aligned}
&\mathbb{E}_{\theta \sim \rho_{\hat{\theta}^{\mathrm{ERM}}}} \left[ r_\theta(\tilde{\mathcal{H}}^S) - R_\theta(\tilde{\mathcal{H}}^S) \right] \\
&\le \frac{D_{\mathrm{kl}}(\rho_{\hat{\theta}^{\mathrm{ERM}}}\|\pi)}{\lambda} + \frac{2\lambda C_2^2 \min\{S,T\}}{(nT)} + \log(4/\delta) && \text{(rearranging terms)} \\
&\le C_2 \sqrt{\min\{S,T\}/(nT)} \cdot \left( D_{\mathrm{kl}}(\rho_{\hat{\theta}^{\mathrm{ERM}}}\|\pi) + 2 \right) + \log(4/\delta) && \text{(by setting } \lambda = \sqrt{nT/\min\{S,T\}} \cdot (1/C_2)) \\
&\le \tilde{\mathcal{O}}(\sqrt{\min\{S,T\}/(nT)}). && \text{(by Lemma E.11)}
\end{aligned} \tag{22}
$$

By Lemma E.12, the loss function is Lipschitz. Since for any $\theta \in \mathrm{supp}(\rho_{\hat{\theta}^{\mathrm{ERM}}})$, $\theta$ is up to $\mathcal{O}(1/(nT))$ away from $\hat{\theta}^{\mathrm{ERM}}$, we can control the difference between the risks of any $\theta \in \mathrm{supp}(\rho_{\hat{\theta}^{\mathrm{ERM}}})$ and $\hat{\theta}^{\mathrm{ERM}}$ as

$$\left| r_\theta(\tilde{\mathcal{H}}^S) - r_{\hat{\theta}^{\mathrm{ERM}}}(\tilde{\mathcal{H}}^S) \right| \le \tilde{\mathcal{O}}(1/(nT)), \tag{23}$$

$$\left| R_\theta - R_{\hat{\theta}^{\mathrm{ERM}}} \right| \le \tilde{\mathcal{O}}(1/(nT)). \tag{24}$$

Thus, we have

$$r_{\hat{\theta}^{\mathrm{ERM}}}(\tilde{\mathcal{H}}^S) - R_{\hat{\theta}^{\mathrm{ERM}}} \le \tilde{\mathcal{O}}(\sqrt{\min\{S,T\}/(nT)}). \tag{25}$$

Applying the above arguments again for the negative of $r$, we have with probability at least $1 - \delta/2$,

$$\left| r_{\hat{\theta}^{\mathrm{ERM}}}(\tilde{\mathcal{H}}^S) - R_{\hat{\theta}^{\mathrm{ERM}}} \right| \le \tilde{\mathcal{O}}(\sqrt{\min\{S,T\}/(nT)}). \tag{26}$$

For $\theta^*$, we can repeat the above steps and get

$$\left| r_{\theta^*}(\tilde{\mathcal{H}}^S) - R_{\theta^*} \right| \le \tilde{\mathcal{O}}(\sqrt{\min\{S,T\}/(nT)}). \tag{27}$$

The probability that all these bounds hold simultaneously is at least $1 - \delta$ w.r.t. $\mathcal{D}$.

Hence with probability at least $1 - \delta$,

$$
\begin{aligned}
&R(\mathtt{TF}_{\hat{\theta}^{\mathrm{ERM}}}) - R(\mathtt{TF}_{\theta^*}) \\
&= R_{\hat{\theta}^{\mathrm{ERM}}} - R_{\theta^*} && \text{(by definition in equation 15)} \\
&\le r_{\hat{\theta}^{\mathrm{ERM}}}(\tilde{\mathcal{H}}^S) - r_{\theta^*}(\tilde{\mathcal{H}}^S) + \tilde{\mathcal{O}}(\sqrt{\min\{S,T\}/(nT)}) && \text{(by equation 26 and equation 27} \\
&= r(\mathtt{TF}_{\hat{\theta}^{\mathrm{ERM}}}) - r(\mathtt{TF}_{\theta^*}) + \tilde{\mathcal{O}}(\sqrt{1/n} + \sqrt{S/T}) && \text{(by definition in equation 14)} \\
&\le \tilde{\mathcal{O}}(\sqrt{\min\{S,T\}/(nT)}) && \text{(by definition of ERM equation 1)}
\end{aligned} \tag{28}
$$

We now take the expectation over each $f^{(i)}$ and $\sigma^{(i)}$ to conclude the proof. $\square$

*Remark* D.1 (Why truncation). Previous analysis (Zhang et al., 2023b) to derive a similar Bayes-optimal argument does not truncate the history and treats the whole history as an inhomogeneous Markov chain. Then they apply the concentration inequalities on Markov chains (for example, Lemma F.2) to control the difference between $R$ and $r$. However, their arguments have two limitations: the first one is that their model is assumed to make decisions based on the full history, which clearly exceeds the Transformer's model's capacity. The second limitation is that such a concentration argument for Markov chains often relies on upper bounding the mixing time or lower bounding the spectral gap (for example, Fan et al. (2021)). But Zhang et al. (2023b) do not specify this the mixing time. Furthermore, in each sampled task sequence (assume we know the task $f^{(i)}$), the mixing time of the (untruncated) history $\tilde{H}_t$ is infinity: if two sequences start with different initial pairs of $(x_1, y_1)$, then they will never become identical no longer what comes consecutively. Thus, their mixing time will be $T$, leading to an $\tilde{O}(1/\sqrt{n})$ generalization, which is suboptimal if $S \ll T$ compared to our result.

# E    Proofs of Lemmas

In this section, we prove these lemmas based on the choice of the activation function $\texttt{act} = \texttt{softmax}$. Similar results for other options $\texttt{act} = \texttt{ReLU}$ can also be found in many existing literatures (for example, see Bai et al. (2024)).

## E.1    Boundedness of Transformers

**Lemma E.1** (Layer-wise boundedness)**.** *Suppose at the l-th layer of the Transformer, we have* $\|W_V^{m,(l)}\|_{2,2} \leq B_V$ *for any* $m = 1, \ldots, M$, $\|A_1^{(l)}\|_{2,2}, \|A_2^{(l)}\|_{2,2} \leq B_A$. *Then for any input* $H^{(l-1)}$, *we have*

$$\|H^{(l)}\|_{2,\infty} \leq (1 + B_A^2)(1 + MB_V)\|H^{(l-1)}\|_{2,\infty}.$$

*Proof of Lemma E.1.* For notation simplicity, we denote $\texttt{softmax}((W_K^{(l)}H^{(l-1)})^\top W_Q^{(l)}H^{(l-1)})$ as $\mathbf{S}^m$. Note that every column of $\mathbf{S}^m$ is of unit 1-norm. Denote each column of $\mathbf{S}^m$ by $s_t^m$. For any input $H$, we have

$$
\begin{aligned}
&\|\texttt{MHA}_{W^{(l)}}(H)\|_{2,\infty} \\
&\leq \|H\|_{2,\infty} + \sum_{m=1}^M \|W_V^{m,(l)}H\mathbf{S}\|_{2,\infty} && \text{(by triangle inequality)} \\
&= \|H\|_{2,\infty} + \sum_{m=1}^M \max_t \|W_V^{m,(l)}Hs_t^m\|_2 && \text{(by definition of } \|\cdot\|_{2,\infty}) \\
&\leq \|H\|_{2,\infty} + \sum_{m=1}^M \max_t \|W_V^{m,(l)}H\|_{2,\infty}\|s_t^m\|_1 && \text{(by Lemma F.5)} \\
&= \|H\|_{2,\infty} + \sum_{m=1}^M \|W_V^{m,(l)}H\|_{2,\infty} && \text{(since } s_t^m \text{ is of unit 1-norm)} \\
&\leq \|H\|_{2,\infty} + \sum_{m=1}^M \|W_V^{m,(l)}\|_{2,2}\|H\|_{2,\infty} && \text{(by Lemma F.6)} \\
&\leq (1 + MB_V)\|H\|_{2,\infty}. && \text{(by assumption of bounded norm)} \quad (29)
\end{aligned}
$$

For any input $H$, we have

$$
\begin{aligned}
&\|\texttt{MLP}_{A^{(l)}}(H)\|_{2,\infty} \\
&\leq \|H\|_{2,\infty} + \|A_2^{(l)}\texttt{ReLU}(A_1^{(l)}H)\|_{2,\infty} && \text{(by triangle inequality)} \\
&\leq \|H\|_{2,\infty} + \|A_2^{(l)}\|_{2,2}\|\texttt{ReLU}(A_1^{(l)}H)\|_{2,\infty} && \text{(by Lemma F.6)} \\
&= \|H\|_{2,\infty} + \|A_2^{(l)}\|_{2,2}\max_t \|\texttt{ReLU}(A_1^{(l)}H)_{:,t}\|_2 && \text{(by definition of } \|\cdot\|_{2,\infty}) \\
&\leq \|H\|_{2,\infty} + \|A_2^{(l)}\|_{2,2}\max_t \|(A_1^{(l)}H)_{:,t}\|_2 && \text{(since } |\texttt{ReLU}(z)| \leq |z| \text{ for any } z \in \mathbb{R}) \\
&= \|H\|_{2,\infty} + \|A_2^{(l)}\|_{2,2}\|A_1^{(l)}H\|_{2,\infty} && \text{(by definition of } \|\cdot\|_{2,\infty}) \\
&\leq \|H\|_{2,\infty} + \|A_2^{(l)}\|_{2,2}\|A_1^{(l)}\|_{2,2}\|H\|_{2,\infty} && \text{(by Lemma F.6)} \\
&\leq (1 + B_A^2)\|H\|_{2,\infty}. && \text{(by assumption of bounded norm)} \quad (30)
\end{aligned}
$$

Combining equation 29 and equation 30 yields the conclusion.   □

**Lemma E.2** (Transformer's boundedness)**.** *Suppose* $\|W_V^{m,(l)}\|_{2,2} \leq B_V$ *for any* $m = 1, \ldots, M$, $\|A_1^{(l)}\|_{2,2}, \|A_2^{(l)}\|_{2,2} \leq B_A$ *for any* $l \in [L]$. *We further assume the projection matrix* $P$ *is of bounded norm* $\|P^\top\|_{2,\infty} \leq B_P$. *Then the Transformer's outputs satisfy that*

$$|\hat{y}(H)| \leq C_1\|H\|_{2,\infty}, \quad and \quad \exp(-C_1\|H\|_{2,\infty}) \leq \hat{\sigma}(H) \leq 1 + C_1\|H\|_{2,\infty},$$

*where $C_1 \coloneqq B_P(1 + B_A^2)^L(1 + MB_V)^L$ is a specified constant.*

*Proof of Lemma E.2.* By Lemma E.1 and a "peeling" argument, we can easily prove that

$$\|H^{(L)}\|_{2,\infty} \le (1 + B_A^2)^L(1 + MB_V)^L\|H^{(0)}\|_{2,\infty}.$$

Thus,

$$\|H_{:,t}^{(L)}\|_2 \le \|H^{(L)}\|_{2,\infty} \le (1 + B_A^2)^L(1 + MB_V)^L\|H^{(0)}\|_{2,\infty}.$$

Denote $P$ by $P = [p_1, p_2]^\top$, where $p_1$ and $p_2$ are vectors of dimension $d$. Then the first output

$$\hat{y} = p_1^\top H_{:,t}^{(L)},$$

where we have (by Cauchy-Schwarz inequality),

$$|\hat{y}| \le \|p_1\|_2\|H_{:,t}^{(L)}\|_2 \le B_P(1 + B_A^2)^L(1 + MB_V)^L\|H^{(0)}\|_{2,\infty}.$$

The other output $\hat{\sigma}$ can be proved similarly as long as one notices

$$\log(1 + \exp(-x)) \ge \exp(-x), \quad \text{and} \quad \log(1 + \exp(x)) \le 1 + x,$$

for any $x \ge 0$. $\square$

**Lemma E.3** (Boundedness of loss)**.** *Under Assumption B.5 with $\|\theta\| \le B_{TF}$ and Assumption B.6 with $\|H\|_{2,\infty} \le B_H$ almost surely, we have*
$$|\ell(\mathit{TF}_\theta(H_t), y_t)| \le C_2$$
*almost surely, where $C_2 \coloneqq (C_1 + 1)^2 B_H^2 \cdot \exp(2C_1 B_H) + \max\{C_1 B_H, 1 + \log(C_1 B_H)\}$ is a specified constant, and $C_1$ is a constant defined in Lemma E.2.*

*Proof of Lemma E.3.* By Lemma E.2, we have

$$\begin{aligned}
\frac{(y_t - \hat{y}(H_t))^2}{2\hat{\sigma}^2(H_t)} &\le (y_t - \hat{y}_t(H_t))^2 \cdot \frac{\exp(2C_1 B_H)}{2} \\
&\le (y_t^2 + \hat{y}_t(H_t)^2) \cdot \exp(2C_1 B_H) \\
&\le (C_1 + 1)^2 B_H^2 \cdot \exp(2C_1 B_H),
\end{aligned}$$

where the second inequality follows from Cauchy's inequality. Combining with a triangle inequality, we have the desired result. $\square$

## E.2 Lipschitzness of Transformers

**Lemma E.4** (Lipschitzness of multi-head attention)**.** *Suppose we define the output's norm as $\|\cdot\|_{2,\infty}$, the norm of $W$ as*
$$\|W\| \coloneqq \max\{\|W_V^m\|_{2,2}, \|W_K^m\|_{2,2}, \|W_Q^m\|_{2,2} : \ m = 1, \ldots, M\},$$

*and the input $H$'s norm as $\|\cdot\|_{2,\infty}$. Suppose at the $l$-th layer of the Transformer, we have $\|W^{m,(l)}\| \le B_W$ for any $m = 1, \ldots, M$, and $\|H^{(l-1)}\|_{2,\infty} \le B_H^{(l-1)}$ almost surely. Then $\mathit{MHA}_{W^{(l)}}(H^{(l-1)})$ is $C_3^{(l)}$-Lipschitz with respect to $W^{(l)}$ and $C_4$-Lipschitz with respect to $H^{(l-1)}$ almost surely. Here $C_3^{(l)} \coloneqq 2B_W^2(B_H^{(l-1)})^3 + (B_H^{(l-1)})$ and $C_4 \coloneqq 1 + MB_W$ are specified constants.*

*Proof of Lemma E.4.* We first prove the Lipschitzness result for $W$. To ease the notations, we omit the dependence on $l$ and sometimes abbreviate $W_K^\top W_Q$ as $W_{KQ}$. For any $W$ and $W'$, using triangle inequality twice, we have

$$\left\|\mathit{MHA}_W(H) - \mathit{MHA}_{W'}(H)\right\|_{2,\infty}$$

$$\leq \sum_{m=1}^{M} \left\| W_V^m H \texttt{softmax}(H^\top W_{KQ}^m H) - W_V^{m\prime} H \texttt{softmax}(H^\top W_{KQ}^{m\prime} H) \right\|_{2,\infty}$$

$$\leq \sum_{m=1}^{M} \left\| W_V^m H \big( \texttt{softmax}(H^\top W_{KQ}^m H) - \texttt{softmax}(H^\top W_{KQ}^{m\prime} H) \big) \right\|_{2,\infty}$$

$$+ \sum_{m=1}^{M} \left\| (W_V^m - W_V^{m\prime}) H \texttt{softmax}(H^\top W_{KQ}^{m\prime} H) \right\|_{2,\infty}. \tag{31}$$

We now deal with two terms in equation 31 separately. Since our conclusion will be made for arbitrary $m \in [M]$, we omit the dependence on $m$ for notation simplicity from now on.

For the first term, we have

$$\left\| W_V H \big( \texttt{softmax}(H^\top W_{KQ} H) - \texttt{softmax}(H^\top W_{KQ}' H) \big) \right\|_{2,\infty}$$

$$= \max_t \left\| W_V H \big( \texttt{softmax}(H^\top W_{KQ} h_t) - \texttt{softmax}(H^\top W_{KQ}' h_t) \big) \right\|_2 \qquad \text{(by definition of } \| \cdot \|_{2,\infty})$$

$$\leq \| W_V H \|_{2,\infty} \cdot \max_t \left\| \big( \texttt{softmax}(H^\top W_{KQ} h_t) - \texttt{softmax}(H^\top W_{KQ}' h_t) \big) \right\|_1 \qquad \text{(by Lemma F.5)}$$

$$\leq \| W_V \|_{2,2} \| H \|_{2,\infty} \cdot \max_t \left\| \big( \texttt{softmax}(H^\top W_{KQ} h_t) - \texttt{softmax}(H^\top W_{KQ}' h_t) \big) \right\|_1 \qquad \text{(by Lemma F.6)}$$

$$\leq 2 \| W_V \|_{2,2} \| H \|_{2,\infty} \cdot \max_t \left\| H^\top W_{KQ} h_t - H^\top W_{KQ}' h_t \right\|_\infty \qquad \text{(by Lemma F.7)}$$

$$\leq 2 \| W_V \|_{2,2} \| H \|_{2,\infty} \cdot \max_t \| H \|_{2,\infty} \left\| W_{KQ} h_t - W_{KQ}' h_t \right\|_2 \qquad \text{(by Lemma F.5)}$$

$$\leq 2 \| W_V \|_{2,2} \| H \|_{2,\infty}^2 \cdot \max_t \left\| W_{KQ} - W_{KQ}' \right\|_{2,2} \| h_t \|_2 \qquad \text{(by Lemma F.5)}$$

$$\leq 2 \| W_V \|_{2,2} \| H \|_{2,\infty}^2 \cdot \left\| W_{KQ} - W_{KQ}' \right\|_{2,2} \| H_t \|_{2,\infty} \qquad \text{((by definition of } \| \cdot \|_{2,\infty})$$

$$\leq 2 \| W_V \|_{2,2} \| H \|_{2,\infty}^3 \cdot \left( \left\| W_K W_Q - W_K W_Q' \right\|_{2,2} + \left\| W_K W_Q' - W_K' W_Q' \right\|_{2,2} \right) \qquad \text{((by triangular inequality)}$$

$$\leq 2 \| W_V \|_{2,2} \| H \|_{2,\infty}^3 \cdot \left( \| W_K \|_{2,2} \| W_Q - W_Q' \|_{2,2} + \| W_K - W_K' \|_{2,2} \| W_Q' \|_{2,2} \right). \qquad \text{((by sub-multiplicativity of matrix norm)}$$

$$\leq 2 B_W^2 (B_H^{(l-1)})^3 \cdot \left( \| W_Q - W_Q' \|_{2,2} + \| W_K - W_K' \|_{2,2} \right). \qquad \text{((by bounded norm assumption)} \tag{32}$$

For notation simplicity, we denote $\texttt{softmax}(H^\top W_{KQ}^{m\prime} H)$ by $\mathtt{S}$. Note that every column of $\mathtt{S}$ is of unit 1-norm. Denote each column of $\mathtt{S}$ by $s_t$. For the second term, we have

$$\left\| (W_V - W_V') H \texttt{softmax}(H^\top W_{KQ}^{m\prime} H) \right\|_{2,\infty}$$

$$= \| (W_V - W_V') H \mathtt{S} \|_{2,\infty} \qquad \text{(by notation substitution)}$$

$$= \max_t \| (W_V - W_V') H s_t \|_2 \qquad \text{(by definition of } \| \cdot \|_{2,\infty})$$

$$\leq \max_t \| (W_V - W_V') H \|_{2,\infty} \| s_t \|_1 \qquad \text{(by Lemma F.5)}$$

$$= \| (W_V - W_V') H \|_{2,\infty} \qquad \text{(since } s_t \text{ is of unit 1-norm)}$$

$$\leq \| W_V - W_V' \|_{2,2} \| H \|_{2,\infty} \qquad \text{(by Lemma F.6)}$$

$$\leq B_H^{(l-1)} \| W_V - W_V' \|_{2,2}. \qquad \text{(by assumption of bounded norm)} \tag{33}$$

Substituting equation 32 and equation 33 into equation 31, we can conclude that $\texttt{MHA}_{W^{(l)}}(H^{(l-1)})$ is $C_3^{(l)}$-Lipschitz with respect to $W^{(l)}$ for $C_3^{(l)} := 2 B_W^2 (B_H^{(l-1)})^3 + (B_H^{(l-1)})$.

As for the second Lipschitzness conclusion (the one w.r.t. $H$), it is straightforward if one replaces $H$ with $H - H'$ in the proof of equation 29. $\qquad \square$

**Lemma E.5** (Lipschitzness of multi-layer perceptron). *Suppose we define the output's norm as $\| \cdot \|_{2,\infty}$, the norm of $W$ as*

$$\| A \| := \max\{ \| A_1 \|_{2,2}, \| A_2 \|_{2,2} \},$$

*and the input $H$'s norm as $\|\cdot\|_{2,\infty}$. Suppose at the $l$-th layer of the Transformer, we have $\|A^{(l)}\| \le B_A$ and $\|H\|_{2,\infty} \le B_H^{\prime(l-1)}$ almost surely. Then $\mathtt{MLP}_{A^{(l)}}(H)$ is $C_5^{(l)}$-Lipschitz with respect to $A^{(l)}$ and $C_6$-Lipschitz with respect to $H$ almost surely. Here $C_5^{(l)} := B_A B_H^{\prime(l-1)}$ and $C_6 := 1 + B_A^2$ are specified constants.*

*Proof of Lemma E.5.* We first prove the Lipschitzness result for $A$. To ease the notations, we omit the dependence on $l$. For any $A$ and $A'$, we have

$$
\begin{aligned}
& \big\|\mathtt{MLP}_A(H) - \mathtt{MLP}_{A'}(H)\big\|_{2,\infty} \\
\le{}& \big\|(A_2 - A_2')\mathtt{ReLU}(A_1 H)\big\|_{2,\infty} + \big\|A_2'(\mathtt{ReLU}(A_1 H) - \mathtt{ReLU}(A_1'))\big\|_{2,\infty} && \text{(by triangle inequality)} \\
\le{}& \|A_2 - A_2'\|_{2,2}\big\|\mathtt{ReLU}(A_1 H)\big\|_{2,\infty} \\
& + \|A_2'\|_{2,2}\big\|\mathtt{ReLU}(A_1 H) - \mathtt{ReLU}(A_1' H)\big\|_{2,\infty} && \text{(by Lemma F.6)} \\
={}& \|A_2 - A_2'\|_{2,2} \max_t \|\mathtt{ReLU}(A_1 H)_{:,t}\|_2 \\
& + \|A_2'\|_{2,2} \max_t \|\mathtt{ReLU}(A_1 H)_{:,t} - \mathtt{ReLU}(A_1' H)_{:,t}\|_2 && \text{(by definition of } \|\cdot\|_{2,\infty}) \\
\le{}& \|A_2 - A_2'\|_{2,2} \max_t \|(A_1 H)_{:,t}\|_2 + \|A_2'\|_{2,2} \max_t \|(A_1 H)_{:,t} - (A_1' H)_{:,t}\|_2 \\
& \text{(since } |\mathtt{ReLU}(z_1) - \mathtt{ReLU}(z_2)| \le |z_1 - z_2| \text{ for any } z_1, z_2 \in \mathbb{R}) \\
={}& \|A_2 - A_2'\|_{2,2}\|A_1 H\|_{2,\infty} + \|A_2'\|_{2,2}\|A_1 H - A_1' H\|_{2,\infty} && \text{(by definition of } \|\cdot\|_{2,\infty}) \\
\le{}& \|A_2 - A_2'\|_{2,2}\|A_1\|_{2,2}\|H\|_{2,\infty} + \|A_2'\|_{2,2}\|A_1 - A_1'\|_{2,2}\|H\|_{2,\infty} && \text{(by Lemma F.6)} \\
\le{}& B_A B_H^{\prime(l-1)}\big(\|A_1 - A_1'\|_{2,2} + \|A_2 - A_2'\|_{2,2}\big) && \text{(by assumption of bounded norm)}
\end{aligned}
$$

$$\tag{34}$$

As for the second Lipschitzness conclusion (the one w.r.t. $H$), it is straightforward if one replaces $H$ with $H - H'$ in the proof of equation 30. $\qquad\square$

**Lemma E.6** (Lipshitzness of Transformer). *Suppose we define each output's norm as $|\cdot|$ for $\hat{y}$ and $\hat{\sigma}$, the norm of $\theta$ as*

$$\|\theta\| := \max\{\|W\|, \|A\|, \|P\|\},$$

*where $\|W\|$ is as defined in Lemma E.4, $\|A\|$ is as defined in Lemma E.5, and $\|P\| := \|P^\top\|_{2,\infty}$, and the input $H$'s norm as $\|\cdot\|_{2,\infty}$. Suppose we have $\|\theta\| \le B_{TF}$, and $\|H\|_{2,\infty} \le B_H$ almost surely. Then $\hat{y}_\theta(H)$ is $C_7$-Lipschitz with respect to $\theta$, and $\hat{\sigma}_\theta(H)$ is $C_8$-Lipschitz with respect to $\theta$.*

*Proof of Lemma E.6.* First we quantify the constants $B_H^{(l-1)}$ in Lemma E.4 and the constants $B_H^{\prime(l-1)}$ in Lemma E.5 via Lemma E.1. As is shown in the proof of Lemma E.1, we can define

$$B_H^{(l-1)} := (1 + M B_{\text{TF}})^{l-1}(1 + B_{\text{TF}}^2)^{l-1}, \quad l = 1, \ldots, L,$$

and

$$B_H^{\prime(l-1)} := (1 + M B_{\text{TF}})^l (1 + B_{\text{TF}}^2)^{l-1}, \quad l = 1, \ldots, L,$$

such that all requirements in Lemma E.4 and Lemma E.5 are met almost surely. Thus, we bound the gap between $H^{(l)}$ (the output of $\mathtt{TF}_\theta$ after $l$ layers) and $H^{\prime(l)}$ (the output of $\mathtt{TF}_{\theta'}$ after $l$ layers) by induction. We claim that if $H^{(0)} = H^{\prime(0)}$, then there exists a constant $C_9^{(l)}$ for any $l = 1, \ldots, L$ that do not depend on $\theta$ or $H$, such that

$$\|H^{(l)} - H^{\prime(l)}\|_{2,\infty} \le C_9^l \|\theta - \theta'\|.$$

We prove it by induction. For $l = 1$, the case can be verified by calculation: by Lemma E.4,

$$\|\mathtt{MHA}_{W^{(1)}}(H^{(0)}) - \mathtt{MHA}_{W'^{(1)}}(H^{(0)})\|_{2,\infty} \le C_3^{(1)}\|\theta - \theta'\|.$$

Similarly, by Lemma E.5,

$$
\begin{aligned}
\|H^{(1)} - H'^{(1)}\|_{2,\infty} &= \|\text{MLP}_{A^{(1)}}(\text{MHA}_{W^{(1)}}(H^{(0)})) - \text{MLP}_{A'^{(1)}}(\text{MHA}_{W'^{(1)}}(H^{(0)}))\|_{2,\infty} \\
&\leq \|\text{MLP}_{A^{(1)}}(\text{MHA}_{W^{(1)}}(H^{(0)})) - \text{MLP}_{A^{(1)}}(\text{MHA}_{W'^{(1)}}(H^{(0)}))\|_{2,\infty} \\
&\quad + \|\text{MLP}_{A^{(1)}}(\text{MHA}_{W'^{(1)}}(H^{(0)})) - \text{MLP}_{A'^{(1)}}(\text{MHA}_{W'^{(1)}}(H^{(0)}))\|_{2,\infty} \\
&\leq C_6\|\text{MHA}_{W^{(1)}}(H^{(0)}) - \text{MHA}_{W'^{(1)}}(H^{(0)})\|_{2,\infty} + C_5^{(1)}\|\theta - \theta'\| \\
&\leq (C_6 C_3^{(1)} + C_5^{(1)})\|\theta - \theta'\|,
\end{aligned}
\tag{35}
$$

where we define $C_9^1$ as $C_9^1 \coloneqq C_6 C_3^{(1)} + C_5^{(1)}$. Suppose our conclusion holds for any $l \leq l_0 - 1$. Then for $l = l_0$, we have

$$
\begin{aligned}
&\|\text{MHA}_{W^{(l_0)}}(H^{(l_0-1)}) - \text{MHA}_{W'^{(l_0)}}(H'^{(l_0-1)})\|_{2,\infty} \\
&\leq \|\text{MHA}_{W^{(l_0)}}(H^{(l_0-1)}) - \text{MHA}_{W^{(l_0)}}(H'^{(l_0-1)})\|_{2,\infty} + \|\text{MHA}_{W^{(l_0)}}(H'^{(l_0-1)}) - \text{MHA}_{W'^{(l_0)}}(H'^{(l_0-1)})\|_{2,\infty} \\
&\leq C_4\|H^{(l_0-1)} - H'^{(l_0-1)}\|_{2,\infty} + C_3^{(l_0)}\|\theta - \theta'\| \\
&\leq (C_4 C_9^{(l_0-1)} + C_3^{(l_0-1)})\|\theta - \theta'\|,
\end{aligned}
$$

by applying Lemma E.4. We can again compute the difference between $H^{(l_0)}$ and $H'^{(l_0)}$ similar to what we do in equation 35 as

$$
\|H^{(l_0)} - H'^{(l_0)}\|_{2,\infty} \leq \left(C_6(C_4 C_9^{(l_0-1)} + C_3^{(l_0-1)}) + C_5^{(l_0)}\right)\|\theta - \theta'\|.
$$

Hence the induction holds if we define $C_9^{(l_0)} \coloneqq C_6(C_4 C_9^{(l_0-1)} + C_3^{(l_0-1)}) + C_5^{(l_0)}$. Now we have proved

$$
\|H^{(L)} - H'^{(L)}\|_{2,\infty} \leq C_9^{(L)}\|\theta - \theta'\|.
$$

We shall see from Cauchy-Schwarz inequality that

$$
\begin{aligned}
|\hat{y} - \hat{y}'| &\leq \|p_1 - p_1'\|_2 \|H^{(L)}\|_{2,\infty} - \|p_1'\|_2 \|H^{(L)} - H'^{(L)}\|_{2,\infty} \\
&\leq \|\theta - \theta'\|(1 + MB_{\text{TF}})^L(1 + B_{\text{TF}}^2)^L + B_{\text{TF}} C_9^{(L)}\|\theta - \theta'\| \\
&= \left((1 + MB_{\text{TF}})^L(1 + B_{\text{TF}}^2)^L + B_{\text{TF}} C_9^{(L)}\right)\|\theta - \theta'\|,
\end{aligned}
$$

where the second inequality follows from the proof of Lemma E.6. We can now define

$$
C_7 \coloneqq (1 + MB_{\text{TF}})^L(1 + B_{\text{TF}}^2)^L + B_{\text{TF}} C_9^{(L)},
$$

and conclude the proof for $\hat{y}$. As for $\hat{\theta}$, we can see from the fact $\log(1 + \exp(\cdot))$ is 1-Lipschitz that the Lipschitzness also holds for $C_8 \coloneqq C_7$. $\qquad\square$

**Lemma E.7** (Lipschitzness of loss). *Suppose we have $\|\theta\| \leq B_{TF}$ and $\|H\|_{2,\infty} \leq B_H$ almost surely, where the norm of $\theta$ is the same as defined in Lemma E.6. Then $\ell(\text{TF}_\theta(H), y)$ is $C_{10}$-Lipschitz with respect to $\theta$ almost surely.*

*Proof of Lemma E.7.* Based on the Lipschitzness of the Transformer w.r.t. $\theta$ (Lemma E.6), we only need to prove that both partial derivatives $\frac{\partial \ell}{\partial \hat{y}}$ and $\frac{\partial \ell}{\partial \hat{\sigma}}$ are bounded. For the first partial derivative, we have

$$
\begin{aligned}
\left|\frac{\partial \ell}{\partial \hat{y}}\right| &= |(y - \hat{y})| \cdot \frac{1}{\hat{\sigma}^2} \\
&\leq (1 + C_1)B_H \exp(2C_1 B_H). \quad \text{(by Lemma E.2)}
\end{aligned}
\tag{36}
$$

For the second partial derivative, we have

$$
\begin{aligned}
\left|\frac{\partial \ell}{\partial \hat{\sigma}}\right| &= \frac{\left|-(y - \hat{y})^2 + \sigma^2\right|}{\hat{\sigma}^3} \\
&\leq \left((1 + C_1)^2 B_H^2 + (1 + C_1 B_H)^2\right) \cdot \exp(3C_1 B_H). \quad \text{(by Lemma E.2)}
\end{aligned}
\tag{37}
$$

Combining inequalities equation 36 and equation 37 with the Lipschitzness of $\hat{y}$ and $\hat{\sigma}$ w.r.t. $\theta$, we conclude the result with

$$C_{10} := (1 + C_1)B_H \exp(2C_1 B_H)C_7 + \left((1 + C_1)^2 B_H^2 + (1 + C_1 B_H)^2\right)\exp(3C_1 B_H)C_8,$$

where $C_7$ and $C_8$ are constants that appear in Lemma E.6. $\qquad\square$

### E.3 Constructing Distributions over Parameter Space

In this section, we formally define two distributions over the parameter space $\Theta$. The first distribution $\rho_{\hat{\theta}}$ may depend on the empirical distribution, while the second distribution $\pi_\theta$ should be independent of the training dataset. We control the Kullback-Leibler divergence between $\rho_{\hat{\theta}}$ and $\pi_\theta$ in Lemma E.11. For notation simplicity, we may use some notations of different meanings from the main text.

For any dimension $d$, we denote the Lebesgue measure over $\mathbb{R}^d$ by $\lambda_d(\cdot)$. Then we have the following lemma.

**Lemma E.8** (Upper bound for p.d.f.)**.** *Suppose $\rho$ is the uniform distribution over $\mathcal{B}(x_0, 3r) \cap \mathcal{B}(0, R)$ for some $x_0 \in \mathcal{B}(0, R) \subset \mathbb{R}^d$, where the Lebesgue measure is defined as $\lambda_d(\cdot)$, and $R > 3r$. Then the p.d.f. $p_\rho(\cdot)$ exists and*

$$p_\rho(x) \leq \frac{1}{\lambda_d\big(\mathcal{B}(0, r)\big)}.$$

*Proof of Lemma E.8.* Denote the set to be $S := \mathcal{B}(x_0, 3r) \cap \mathcal{B}(0, R)$. Since $\rho$ is the uniform distribution, we just need to prove that

$$\lambda_d(S) \geq \lambda_d\big(\mathcal{B}(0, r)\big).$$

This is true because there exists some $x' \in \mathbb{R}^d$ s.t. $\mathcal{B}(x', r) \subset S$. In fact, we can construct the small ball as

$$\mathcal{B}\Big(x_0 - \frac{x_0}{\|x_0\|} \cdot 1.5r, r\Big) \subset S.$$

$\qquad\square$

**Lemma E.9** (Upper bound for KL divergence)**.** *Suppose the probability space is defined on $\mathcal{B}(0, R)$. Suppose $\rho$ is the uniform distribution over $\mathcal{B}(x_0, 3r) \cap \mathcal{B}(0, R)$ for some $x_0 \in \mathcal{B}(0, R) \subset \mathbb{R}^d$, where the Lebesgue measure is defined as $\lambda_d(\cdot)$, and $R > 3r$. Suppose $\pi$ is the uniform distribution over $\mathcal{B}(0, R)$. Then*

$$D_{\mathrm{kl}}(\rho\|\pi) \leq \mathcal{O}(C_d \cdot \log(R/r)),$$

*where $C_d := \log(\lambda_d(\mathcal{B}(0, 1)))$ is some constant related to $d$.*

*Proof of Lemma E.9.* Since $\rho \ll \pi$, we can define the Radon-Nikodym derivative as $\frac{\mathrm{d}\rho}{\mathrm{d}\pi}$. By Lemma E.8, we can upper bound the RN derivative by

$$\frac{\mathrm{d}\rho}{\mathrm{d}\pi}(x) = \frac{1/\lambda_d(\mathcal{B}(x_0, 3r) \cap \mathcal{B}(0, R))}{1/\lambda_d\mathcal{B}(0, R)} \leq \mathcal{O}(C_d \cdot \log(R/r)).$$

Hence,

$$\begin{aligned}
D_{\mathrm{kl}}(\rho\|\pi) &= \int_{x \in \mathcal{B}(0, R)} \log\Big(\frac{\mathrm{d}\rho}{\mathrm{d}\pi}(x)\Big)\mathrm{d}\rho(x) \\
&\leq \int_{x \in \mathcal{B}(0, R)} \mathcal{O}(C_d \cdot \log(R/r))\mathrm{d}\rho(x) \\
&= \mathcal{O}(C_d \cdot \log(R/r)).
\end{aligned}$$

$\qquad\square$

*Remark E.10.* Note that $C_d = \frac{\pi^{\frac{n}{2}}}{\Gamma(\frac{n}{2}+1)}$ is uniformly upper bounded. Here $\pi$ denotes the ratio of a circle's circumference to its diameter, and $\Gamma$ is the Gamma-function.

**Lemma E.11** (Upper bound for $D_{\mathrm{kl}}(\rho_{\hat{\theta}}\|\pi_\theta)$). *Suppose we are considering probability measures over the space specified by Assumption B.6 (that is, $\Theta = \mathcal{B}(0, B_{TF})$). For each layer $l$ and each $m$, suppose we define the norm over each $W_Q, W_K \in \mathbb{R}^{d_m \times d}$, $W_V \in \mathbb{R}^{d \times d}$, $A_1 \in \mathbb{R}^{d_h \times d}$, $A_2 \in \mathbb{R}^{d \times d_h}$ to be the Frobenius norm (that is, $\|\cdot\|_{2,2}$). Suppose $P = [p_1, p_2]^\top$, and we define the norm over $p_1, p_2 \in \mathcal{R}^d$ to be the Euclidean norm. For each layer $l$ and each $m$, suppose we have the probability measures $\rho_{\hat{W}_Q}$, $\rho_{\hat{W}_K}$, $\rho_{\hat{W}_V}$, $\rho_{\hat{A}_1}$, $\rho_{\hat{A}_2}$ as the uniform distribution over $\mathcal{B}(0, 1/(nT)) \cap \mathcal{B}(0, B_{TF}))$, and the probability measures $\pi_{W_Q}$, $\pi_{W_K}$, $\pi_{W_V}$, $\pi_{A_1}$, $\pi_{A_2}$ as the uniform distribution over $\mathcal{B}(0, B_{TF}))$. Suppose we have the probability measures $\rho_{\hat{p}_1}$, $\rho_{\hat{p}_2}$ as the uniform distribution over $\mathcal{B}(0, 1/(nT)) \cap \mathcal{B}(0, B_{TF}))$, and the probability measures $\pi_{p_1}$, $\pi_{p_2}$ as the uniform distribution over $\mathcal{B}(0, B_{TF}))$. Suppose we define*

$$\rho_{\hat{\theta}} := \Big(\bigotimes_{m,l} \rho_{\hat{W}_Q^{m,(l)}}\Big) \otimes \Big(\bigotimes_{m,l} \rho_{\hat{W}_K^{m,(l)}}\Big) \otimes \Big(\bigotimes_{m,l} \rho_{\hat{W}_V^{m,(l)}}\Big)$$
$$\otimes \Big(\bigotimes_l \rho_{\hat{A}_1^{(l)}}\Big) \otimes \Big(\bigotimes_l \rho_{\hat{A}_2^{(l)}}\Big)$$
$$\otimes \rho_{\hat{p}_1} \otimes \rho_{\hat{p}_2}, \tag{38}$$

*and*

$$\pi_\theta := \Big(\bigotimes_{m,l} \pi_{W_Q^{m,(l)}}\Big) \otimes \Big(\bigotimes_{m,l} \pi_{W_K^{m,(l)}}\Big) \otimes \Big(\bigotimes_{m,l} \pi_{W_V^{m,(l)}}\Big)$$
$$\otimes \Big(\bigotimes_l \pi_{A_1^{(l)}}\Big) \otimes \Big(\bigotimes_l \pi_{A_2^{(l)}}\Big)$$
$$\otimes \pi_{p_1} \otimes \pi_{p_2}, \tag{39}$$

*where $\otimes$ represents the product of measures. Then we have*

$$D_{\mathrm{kl}}(\rho_{\hat{\theta}}\|\pi_\theta) \leq \mathcal{O}\big(C_{11}\log(nTB_{TF})\big),$$

*where $C_{11}$ is some specified constant that depends polynomially on $L, M, d, d_m, d_h$.*

*Proof of Lemma E.11.* By setting $r = \frac{1}{3nT}$ for Lemma E.9 and $R = B_{\mathrm{TF}}$, we have for each $m = 1, \ldots, M$ and $l = 1, \ldots, L$,

$$D_{\mathrm{kl}}(\rho_{\hat{W}_Q^{m,(l)}}\|\pi_{W_Q^{m,(l)}}) \leq \mathcal{O}\big(C_{dd_m}\log(nTB_{\mathrm{TF}})\big),$$
$$D_{\mathrm{kl}}(\rho_{\hat{W}_K^{m,(l)}}\|\pi_{W_K^{m,(l)}}) \leq \mathcal{O}\big(C_{dd_m}\log(nTB_{\mathrm{TF}})\big),$$
$$D_{\mathrm{kl}}(\rho_{\hat{W}_V^{m,(l)}}\|\pi_{W_V^{m,(l)}}) \leq \mathcal{O}\big(C_{d^2}\log(nTB_{\mathrm{TF}})\big),$$
$$D_{\mathrm{kl}}(\rho_{\hat{A}_1^{(l)}}\|\pi_{A_1^{(l)}}) \leq \mathcal{O}\big(C_{dd_h}\log(nTB_{\mathrm{TF}})\big),$$
$$D_{\mathrm{kl}}(\rho_{\hat{A}_2^{(l)}}\|\pi_{A_2^{(l)}}) \leq \mathcal{O}\big(C_{dd_h}\log(nTB_{\mathrm{TF}})\big),$$
$$D_{\mathrm{kl}}(\rho_{\hat{p}_1}\|\pi_{p_1}) \leq \mathcal{O}\big(C_d\log(nTB_{\mathrm{TF}})\big),$$
$$D_{\mathrm{kl}}(\rho_{\hat{p}_2}\|\pi_{p_2}) \leq \mathcal{O}\big(C_d\log(nTB_{\mathrm{TF}})\big).$$

By Lemma F.8, we can sum up the above inequalities and get the final result. $\qquad\square$

**Lemma E.12** (Bounded difference). *For any $\hat{\theta} \in \Theta$, suppose we construct the distribution $\rho_{\hat{\theta}}$ as in equation 38. Then for any $\theta \in \mathrm{supp}(\rho_{\hat{\theta}})$, under Assumption B.6 and Assumption B.5, we have*

$$\Big|\ell(\mathit{TF}_\theta(H), y) - \ell(\mathit{TF}_{\hat{\theta}}(H), y)\Big| \leq \mathcal{O}\big(C_{10}/(nT)\big),$$

*almost surely. Here $C_{10}$ is the same as defined in Lemma E.7.*

*Proof of Lemma E.12.* By construction shown in equation 38, we can see that for any $\theta \in \mathrm{supp}(\rho_{\hat{\theta}})$,

$$\|\theta - \hat{\theta}\| \leq 1/(nT).$$

Then from the Lipschitzness of the loss function w.r.t. $\theta$ (Lemma E.7), we conclude the proof. $\qquad\square$

### E.4 Markov Chain's Property

**Lemma E.13** ($\tilde{\mathcal{H}}^S$ is a Markov chain (conditioned on knowing $f$ and $\sigma$))**.** *Suppose we have $\tilde{\mathcal{H}}^S$ defined as equation 13. Then $\tilde{\mathcal{H}}^S$ is a Markov chain conditioned on knowing each $f^{(i)}$ and $\sigma^{(i)}$ for each $i = 1, \ldots, n$.*

*Proof of Lemma E.13.* By definition, the state of $\tilde{\mathcal{H}}^S$ will restart and does not depend on all previous histories once $\tilde{H}_k^S$'s index $k$ reaches the point of $k = iT + 1$. Therefore, we only need to verify that inside each task's sequence, the state $\tilde{H}_k^S$ is also Markovian.

Suppose $k = iT + t$ for some $i$, and we considering $k = iT + 1, \ldots, iT + T$ for each $t = 1, \ldots, T$. We write $\tilde{H}_k^S$ and $(x_{\max\{1, t-S\}}, y_{\max\{1, t-S\}}, \ldots, x_t, y_t)$ interchangeably for notation simplicity.

Each pair of $(x_t, y_t)$ is now independent conditioned on knowing the underlying $f^{(i)}$ and $\sigma^{(i)}$. We omit the conditional dependencies on $f^{(i)}$ and $\sigma^{(i)}$ for notation simplicity. The p.d.f. of $\tilde{H}_k^S$ conditioned on observing $\{\tilde{H}_\tau^S\}_{\tau=iT+1}^{iT+t}$ and knowing $f^{(i)}$ and $\sigma^{(i)}$ is

$$
\begin{aligned}
& p(x_{\max\{1, t-S\}}, y_{\max\{1, t-S\}}, \ldots, x_t, y_t | \{\tilde{H}_\tau^S\}_{\tau=iT+1}^{iT+t} = \{\tilde{H}_\tau^{S\prime}\}_{\tau=iT+1}^{iT+t}, f = f^{(i)}, \sigma = \sigma^{(i)}) \\
&= \mathbb{1}\{x_{\max\{1, t-S\}} = x'_{\max\{1, t-S\}}, \ldots, y_{t-1} = y'_{t-1}\} \cdot p(x_t, y_t | f = f^{(i)}, \sigma = \sigma^{(i)}) \\
& \quad \text{(by conditional independence of each pair of } (x_\tau, y_\tau)) \\
&= p(x_{\max\{1, t-S\}}, y_{\max\{1, t-S\}}, \ldots, x_t, y_t | \tilde{H}_{t-1}^S = \tilde{H}_{t-1}^{S\prime}, f = f^{(i)}, \sigma = \sigma^{(i)})
\end{aligned}
$$

Thus the Markovian property holds. $\qquad\square$

We present the definition of *mixing time* as used in Paulin (2015).

**Definition E.14** (Mixing time for inhomogeneous Markov chains)**.** Let $X_1, \ldots, X_N$ be a Markov chain with Polish state space $\Omega_1 \times \cdots \times \Omega_N$ (that is, $X_i \in \Omega_i$). Let $\mathcal{L}(X_{i+t} | X_i = x)$ be the conditional distribution of $X_{i+t}$ given $X_i = x$. Let us denote the minimal $t$ such that $\mathcal{L}(X_{i+t} | X_i = x)$ and $\mathcal{L}(X_{i+t} | X_i = y)$ are less than $\epsilon$ away in total variational distance for every $1 \leq i \leq N - t$ and $x, y \in \Omega_i$ by $\tau(\epsilon)$, that is, for $0 < \epsilon < 1$, let

$$
\begin{aligned}
\bar{d}(t) &:= \max_{1 \leq i \leq N-t} \sup_{x,y \in \Omega_i} \mathrm{TV}(\mathcal{L}(X_{i+t} | X_i = x), \mathcal{L}(X_{i+t} | X_i = y)), \\
\tau(\epsilon) &:= \min\{t \in \mathbb{N} \; : \; \bar{d}(t) \leq \epsilon\}.
\end{aligned}
$$

We now upper bound the mixing time of $(H_t^S, y_t)$.

**Lemma E.15** (Mixing time for truncated history)**.** *Suppose we are considering the conditional distribution on knowing each $f^{(i)}$ and $\sigma^{(i)}$. Then for the Markov chain $\tilde{\mathcal{H}}_k^S$, we have*

$$
\tau(\epsilon) \leq \min\{S, T\},
$$

*for any $\epsilon \in [0, 1)$.*

*Proof of Lemma E.15.* We first consider the case when $S \leq T$. The mixing property inside each sequence $\tilde{H}_k^S$ for $k = iT + 1, \ldots, iT + T$. Since each $(x, y)$ is i.i.d. distributed conditioned on knowing $f^{(i)}$ and $\sigma^{(i)}$, the conditional distribution of the consecutive sequence $(x_{t+1}, y_{t+1}), \ldots, (x_T, y_T)$ is never affected by previous $t$ pairs $(x_1, y_1), \ldots, (x_t, y_t)$ for any $1 \leq t \leq T$. We consider the conditional distribution on knowing $f^{(i)}$ and $\sigma^{(i)}$ from now on and omit the dependencies for notation simplicity.

For any $1 \leq t \leq T - S$, for any two points $\tilde{H}_{iT+t}^{S\prime} \neq \tilde{H}_{iT+t}^{S\prime\prime}$, the distribution of $\tilde{H}_{iT+t+S}^S$ is independent of previous $t$ pairs of observed samples. In other words,

$$
\mathcal{L}(\tilde{H}_{iT+t+t'}^S | \tilde{H}_{iT+t}^S = \tilde{H}_{iT+t}^{S\prime}) = \mathcal{L}(\tilde{H}_{iT+t+t'}^S | \tilde{H}_{iT+t}^S = \tilde{H}_{iT+t}^{S\prime\prime}),
$$

for any $t' \geq S$. Hence,

$$
\bar{d}(t) = 0, \quad \text{for any } t \geq S.
$$

We have

$$\tau(\epsilon) \leq S, \quad \text{for any } \epsilon \in [0, 1).$$

When $S > T$, note that the flattened (truncated) history $\tilde{H}_k^S$ restarts every time it meets the end of a sequence generated by some $f^{(i)}$ and $\sigma^{(i)}$. Since the length of those sequences is $T$, we have

$$\tau(\epsilon) \leq T, \quad \text{for any } \epsilon \in [0, 1).$$

$\square$

## F   Technical Lemmas

In this section, we present some technical lemmas. Note that all the notations in this section are chosen for simplicity and may have different meanings than those in other sections.

**Lemma F.1** (McDiarmid's inequality (McDiarmid et al., 1989)). *Let $X = (X_1, \ldots, X_N)$ be a vector of independent random variables taking values in a Polish space $\Lambda = \Lambda_1 \times \cdots \times \Lambda_N$. Suppose that $f \colon \Lambda \to \mathbb{R}$ satisfies*

$$f(x) - f(y) \leq \sum_{i=1}^N c_i \mathbb{1}\{x_i \neq y_i\},$$

*for any $x, y \in \Lambda$. Then for any $\lambda \in \mathbb{R}$,*

$$\mathbb{E}\Big[ \exp\big( \lambda(f(X) - \mathbb{E}[f(X)]) \big) \Big] \leq \frac{\lambda^2 \|c\|_2^2}{2}.$$

**Lemma F.2** (Corollary 2.11 in Paulin (2015)). *Let $X = (X_1, \ldots, X_N)$ be a Markov chain taking values in a Polish space $\Lambda = \Lambda_1 \times \cdots \times \Lambda_N$, with mixing time $\tau(\epsilon)$ for $0 \leq \epsilon < 1$. Define*

$$\tau_{\min} \coloneqq \inf_{\epsilon \in [0,1)} \tau(\epsilon) \cdot \Big( \frac{2 - \epsilon}{1 - \epsilon} \Big)^2.$$

*Suppose that $f \colon \Lambda \to \mathbb{R}$ satisfies*

$$f(x) - f(y) \leq \sum_{i=1}^N c_i \mathbb{1}\{x_i \neq y_i\},$$

*for any $x, y \in \Lambda$. Then for any $\lambda \in \mathbb{R}$,*

$$\mathbb{E}\Big[ \exp\big( \lambda(f(X) - \mathbb{E}[f(X)]) \big) \Big] \leq \frac{\lambda^2 \tau_{\min} \|c\|_2^2}{8}.$$

**Lemma F.3** (Donsker-Varadhan variational formula (Donsker & Varadhan, 1983)). *Let $P$ and $Q$ be two probability distributions over $(\Theta, \mathcal{F})$. If $Q \ll P$, then for any real-valued function $h$ integrable w.r.t. $P$,*

$$\log \mathbb{E}_P[\exp h] = \sup_{Q \ll P} \{ \mathbb{E}_Q[h] - D_{\mathrm{kl}}(Q \| P) \}.$$

**Lemma F.4** (Chernoff's bound (Chernoff, 1952)). *For any random variable $X$, if $\mathbb{E}[\exp(X)] \leq 1$, then for any $\delta \in (0, 1)$,*

$$\mathbb{P}(X \leq \log(1/\delta)) \geq 1 - \delta.$$

**Lemma F.5** (Lemma M.7 in Zhang et al. (2022)). *Given any two conjugate numbers $p, q \in [1, \infty]$ s.t. $1/p + 1/q = 1$, for any $r \in [1, \infty]$, we have*

$$\|Ax\|_r \leq \|A\|_{r,p} \|x\|_q, \quad \text{and} \quad \|Ax\|_r \leq \|A^\top\|_{p,r} \|x\|_q$$

*for any matrix $A \in \mathbb{R}^{m \times n}$ and vector $x \in \mathbb{R}^n$.*

**Lemma F.6** (Lemma M.8 in Zhang et al. (2022)). *Given any two conjugate numbers $p, q \in [1, \infty]$ s.t. $1/p + 1/q = 1$, we have*

$$\|AB\|_{p,\infty} \leq \|A\|_{p,q} \|B\|_{p,\infty}$$

*for any matrix $A \in \mathbb{R}^{m \times n}$ and matrix $B \in \mathbb{R}^{n \times r}$.*

**Lemma F.7** (Lemma M.9 in Zhang et al. (2022)). *Given any two vectors $x, y \in \mathcal{R}^d$, we have*

$$\|\text{softmax}(x) - \text{softmax}(y)\|_1 \leq 2\|x - y\|_\infty.$$

**Lemma F.8** (Property of Kullback-Leibler divergence, Proposition 7.2 in Polyanskiy & Wu (2024)). *Given any two probability distributions $\mu_1$ and $\mu_2$ over $(\Omega, \mathcal{F})$ and any two distributions $\nu_1$ and $\nu_1$ over $(\Omega', \mathcal{F}')$, if $\mu_1 \ll \mu_2$ and $\nu_1 \ll \nu_2$, then we have*

$$D_{\text{kl}}(\mu_1 \otimes \nu_1 \| \mu_2 \otimes \nu_2) = D_{\text{kl}}(\mu_1 \| \mu_2) + D_{\text{kl}}(\nu_1 \| \nu_2).$$

# G Experiment Details

## G.1 Training data generation

We first describe a basic setup of all our experiments. For some experiments, we change some part(s) in below to design the corresponding "flipped" experiment or to examine the OOD ability of the trained transformer. In particular, the $i$-th thread the training data

$$\left( x_1^{(i)}, y_1^{(i)}, x_2^{(i)}, y_2^{(i)}, ..., x_T^{(i)}, y_T^{(i)} \right)$$

is generated by the following distributions:

- $\mathcal{P}_X$: the feature vector $x_t^{(i)} \overset{\text{i.i.d.}}{\sim} \mathcal{N}(0, I_d)$ where $I_d$ is $d$-dimensional identity matrix.

- $\mathcal{P}_\epsilon$: the noise $\epsilon_t^{(i)} \overset{\text{i.i.d.}}{\sim} \mathcal{N}(0, 1)$.

- $\mathcal{P}_\sigma$: the noise intensity $\sigma_i$ is sampled i.i.d. from

$$\tau_i \sim \text{Gamma}(\underline{\tau}, \bar{\tau}), \quad \sigma_i = \frac{1}{\sqrt{\tau_i}}$$

  where the parameters $\underline{\tau} = \bar{\tau} = 20$ for the basic setup of the experiment. We change these two parameters for some OOD experiments.

- $\mathcal{P}_\mathcal{F}$: The function $f_i(x) := w_i^\top x$ where $w_i$ is generated from

$$w_i | \sigma_i \sim \mathcal{N}(\bar{w}, \sigma_i^2 \cdot I_d)$$

  where $I_d$ is the $d$-dimension identity matrix and $\bar{w}$ is set to be an all-one vector of dimension $d$. The covariance matrix of $w_i$ is related with the noise intensity $\sigma_i$ to control the signal-to-noise ratio.

Finally, the target variable is calculated by

$$y_t^{(i)} = w_i^\top x_t^{(i)} + \sigma_i \epsilon_t^{(i)}.$$

Throughout the paper, we consider the dimension $d = 8$.

## G.2 Number of Tasks $N$ and Training Procedure

In the previous Section G.1, we define how we generate the training data. As in the previous work, we introduce the notion of *task* where each realization of $(w_i, \sigma_i)$ is referred to as one task. The rationale is that each configuration of $(w_i, \sigma_i)$ corresponds to one pattern of the sequence $(x_t, y_t)$'s. While the distribution of $(w_i, \sigma_i)$ corresponds to infinitely many possible task configurations, we use a finite pool of tasks for training the Transformer. Specifically, we generate

$$\mathcal{T} := \{(w_i, \sigma_i)\}_{i=1}^N$$

from the distributions discussed above. Throughout the paper, we use $N$ to refer to the total number of tasks or the pool size.

Training the Transformer for our setting is slightly different from the classic ML model's training. We do not use a fixed set of training data. Rather, we generate a new batch of training data freshly for each batch.

- The batch size $b = 64$. For each batch, we first sample with replacement $b$ tasks from the task pool $\mathcal{T}$. And based on each sampled $(w_i, \sigma_i)$, we generate a training sequence $\left(x_1^{(i)}, y_1^{(i)}, x_2^{(i)}, y_2^{(i)}, ..., x_T^{(i)}, y_T^{(i)}\right)$ following the setting in the previous Section G.1.

- All the numerical experiments in our paper run for 200,000 batches.

The validation and testing sets are also randomly generated instead of fixed beforehand. But unlike the training phase which draws the task configuration from the task pool $\mathcal{T}$, the validation and test phase samples $(w_i, \sigma_i)$ directly from the original distribution described in the previous Section G.1. This is aimed to validate or test whether the trained model has learned the ability to solve *a family of* problems, or it only just memorizes a fixed pool of tasks $\mathcal{T}$.

## G.3 Derivation of Bayes-optimal Predictor

In Proposition 3.1, we state the Bayes-optimal predictor in the form of a posterior expectation. Now we calculate the Bayes-optimal predictor explicitly under the generation mechanism specified in Section G.1. Conditional on history $H_t = (x_1, y_1, \ldots, x_t)$, the posterior distribution of $(w, \sigma)$ that governs the generation of $H_t$ can be calculated based on the Bayesian posterior as

$$\mathbb{P}(\tau|H_t) = \mathrm{Gamma}(\tau; \underline{\tau}_t, \bar{\tau}_t), \quad \sigma = \frac{1}{\sqrt{\tau}},$$

$$\mathbb{P}(w|\sigma, H_t) = \mathcal{N}(w_t, \sigma^2 \cdot \Sigma_t),$$

where

$$\Sigma_t = \left(I_d + \sum_{s=1}^{t-1} x_s x_s^\top\right)^{-1}, \qquad w_t = \Sigma_t \left(\bar{w} + \sum_{s=1}^{t-1} x_s y_s\right)$$

$$\underline{\tau}_t = \underline{\tau} + \frac{t}{2}, \qquad\qquad \bar{\tau}_t = \bar{\tau} + \frac{1}{2}\sum_{s=1}^{t-1} \left(y_s^2 + \bar{w}^\top \bar{w} - w_s^\top \Sigma_t^{-1} w_s\right).$$

Accordingly, the Bayes-optimal predictor becomes

$$y_t^*(H_t) = \mathbb{E}[y_t|H_t] = w_t^\top x_t,$$

and

$$\sigma_t^{*2}(H_t) = \mathbb{E}[(y_t - y_t^*(H_t))^2|H_t] = \mathbb{E}[(f(x_t) - y_t^*(H_t))^2|H_t] + \mathbb{E}[\sigma^2|H_t]$$

$$= \frac{\bar{\tau}_t}{\underline{\tau}_t - 1} \cdot \left(\mathrm{tr}\left(x_t x_t^\top \Sigma_t\right) + 1\right).$$

These formulas are used to generate the Bayes-optimal curves in the figures.

