# OpenReview forum: "Towards Better Understanding of In-Context Learning Ability from In-Context Uncertainty Quantification"
_TMLR — Accepted by TMLR_

### Review · Reviewer_corT · 2025-02-21

**Summary Of Contributions:**

The authors revisit the training of Transformers on linear regression tasks: they consider a bi-objective prediction task of predicting both the conditional expectation and the conditional variance.  This additional uncertainty quantification objective allows the authors to (i) better design out-of-distribution experiments to distinguish in-context learning from in-weight learning (IWL) and (ii) make a better separation between the algorithms with and without using the prior information of the training distribution. Their theoretical findings are corroborated by compelling empirical experiments.

**Audience:**

Yes

**Broader Impact Concerns:**

There are no concerns on the ethical implications of the work.

**Claims And Evidence:**

Yes

**Requested Changes:**

**Q1** Is it always guaranteed that the Bayes-optimal predictor in Def 2.1 is unique? Is it assumed? If not, can the authors prove it? If not, it'd be enough to substitute $:=$ with $\in$

**Q2** The problem setup considered by the authors is similar to the one put forth in https://openreview.net/forum?id=AH5KwUSsln. It would be nice to see a discussion on how they relate (even in Appendix A). More in general, credal sets or second-order distributions may be of interest for the authors when modelling distribution misspecification or drift.

**Q3** The authors discuss epistemic uncertainty, and measure it via conditional expectation. The latter has been abundantly shown to be a suboptimal way of quantifying EU (see e.g. https://openreview.net/forum?id=4NHF9AC5ui, Section 2.1). While we agree with the authors that it could be computationally preferable to work with such a measure in the context of LLMs, we at least encourage them to add a discussion on EU measures involving credal sets and second-order distributions (arguably, the most principled way of quantifying EU) by referencing works such as https://proceedings.mlr.press/v216/sale23a/sale23a.pdf, https://proceedings.mlr.press/v216/wimmer23a.html, https://openreview.net/forum?id=MhLnSoWp3p, https://openreview.net/forum?id=1Wg0J2WtEU, https://proceedings.mlr.press/v235/sale24a.html, https://arxiv.org/abs/2401.00276, https://arxiv.org/abs/2308.14815

**Q4** Am I correct if I understood that the experiments in Sec 4.1 are somehow similar to those in https://openreview.net/forum?id=4NHF9AC5ui, Section 4.2?

**Q5** Does the in-context learning/in-weight learning problems discussed by the authors relate to continual and/or active learning (see e.g. https://arxiv.org/abs/2308.14815, https://arxiv.org/abs/2305.14782)?

**Strengths And Weaknesses:**

The paper is generally well-written, and the theoretical claims are largely correct. I would recommend for an acceptance after the requested changes (mostly dealing with links to existing literature) are implemented.

Theorem 3.2 is a really nice one.

---

> ### Author Response · Authors · 2025-03-17
>
> We thank the reviewer for appreciating our work and the precious suggestions.
>
> **Q1** We have revised Definition 2.1 accordingly. Thanks for pointing it out.
>
> **Q2** We have added some discussions in Appendix A. Thanks for your notice, yet there is a gap between the purposes. We are not considering proving some generalization bound for the Transformers when facing distribution drift/misspecification. We are just trying to show that the *Bayes optimality* comes naturally from the current ERM approach of studying ICL; it doesn't imply that the Transformers are doing it in a *Bayesian* way. Our theoretical result (e.g. Theorem 3.2) is given in an ID setting. That's different from domain adaptation/generalization or credal set learning theory.
>
> **Q3** We have added some discussions in Appendix A. In short, the ICL ability of Transformers has been a research topic since the so-called zero-shot learning ability was observed in Transformer-based models. Different researchers designed various function classes to examine the dynamics of such interesting phenomena. In this paper, we are designing simple yet illustrative uncertainty quantification examples to test the ICL ability of Transformers rather than proving Transformers should be (one of) the most appropriate approach(es) to do uncertainty quantification. In that light, our work differs from the aim of the above-mentioned uncertainty quantification literature. Although we factorized the uncertainty into two parts in Proposition 3.1, we didn't try to manage each term accordingly (the AU and the EU). Yet we still thank the reviewer for bringing up those works.
>
> **Q4** Yes. The experiments are similar in these aspects: designing different levels of noise, testing in both ID and OOD settings, and trying to quantify the uncertainty. However, the spirit of the experiment is different. As mentioned before, our work is to examine what the Transformers would predict when facing ID/OOD data rather than proving the superiority of Transformers in the task of UQ. Thus, we design several easy-to-check samples, while *Credal Bayesian Deep Learning* tests the UQ ability of their method on some real datasets.
>
> **Q5** Thanks for your interesting question, yet we don't think the ICL ability has a direct relation with continual learning (CL) or active learning (AL). The biggest difference might be that both CL and AL update the model (e.g. parameters) when seeing some new examples, while the ICL or the zero-shot learning doesn't modify the model parameters when facing a sequence of newly arrived examples. It is interesting to study if CL or AL can be adapted to the training of Transformers (e.g. facing different batches of requests, or intentionally deciding which data to annotate), but that's beyond the scope of this work.

---

> > ### Comment · Reviewer_corT · 2025-03-17
> >
> > Thanks for your detailed answers. I am happy to recommend for acceptance of the work.

---

### Review · Reviewer_zqB3 · 2025-03-04

**Summary Of Contributions:**

1. This work theoretically proves that, for i.i.d. scenarios, when a nonlinear Transformer is trained on ICL data, the downstream ICL performance will converge to a population risk minimizer given long enough context.
2. This work investigates three OOD generalization scenarios: task shift, covariate shift (input distribution shift), and length shift. It empirically reveals that, for task shift, the ICL prediction deviates from Bayesian-optimal predictions; for covariate shift, training on a larger meta-distribution can improve generalization; for length shift, it is mainly caused by the distributional shift in the positional encodings.

**Audience:**

Yes

**Broader Impact Concerns:**

No concerns on the ethical implications.

**Claims And Evidence:**

Yes

**Requested Changes:**

1. Add more discussions about how the practical value of the theory.
2. In Section4.1, in the sentence "Comparatively, the mere observation that the Transformer works well on the in-distribution data does not demonstrate its in-context learning ability as a traditional supervised learning model also has such ability and generalization performance over in-distribution data", should "... generalization performance over in-distribution data" be "... generalization performance over OOD data"? Is this a typo?

**Strengths And Weaknesses:**

**Strengths**

1. The theory considers a very general multi-layer Transformer architecture with multi-head nonlinear activations, tackling a challenging theoretical setting. The limited context window design aligns well with practical scenarios.
2. The paper provides comprehensive and detailed comparison with previous theoretical literature, making it clear to understand the theoretical contributions of this work.
3. The findings in Sec. 4.3 that the failure in length generalization of ICL is mainly due to the distributional shift in the positional encodings is insightful.

**Weaknesses**

1. The main theoretical finding of Theorem 3.2 that ICL converges to a Bayesian-optimal predictor lacks novel insights. Many previous works [1-2] have empirically and theoretically revealed that ICL achieve nearly optimal performances on the i.i.d. linear regression task, and the performance improves with longer context length and more training examples.
2. The practical value of the theory remains unclear. Although the theory considers a more general Transformer architecture and provides a tighter generalization bound, I'm still concerned about the practical value of the theory. Specifically, what does a tighter generalization bound imply for real-world applications? How can these theoretical insights inform the design of more effective ICL algorithms or paradigms?
3. The results in Sec. 4.1 is somewhat less surprising, since previous works [1-3] have highlighted the limited OOD performance of ICL. Similarly, the result in Sec. 4.2 also seems not surprising to me either, since in my opinion, the ICL performance will get better as the test distribution becomes closer to the training one. Given that the proposed meta-distribution training enlarges the training distribution, it's plain to infer that it will help generalize to new distributions.

[1] What Can Transformers Learn In-Context? A Case Study of Simple Function Classes, NeurIPS 2022

[2] Trained transformers learn linear models in-context, JMLR 2023

[3] A Closer Look at In-Context Learning under Distribution Shifts, Arxiv 2023.05

[4] Pretraining Data Mixtures Enable Narrow Model Selection Capabilities in Transformer Models, Arxiv 2023.11

[5] Can in-context learning really generalize to out-of-distribution tasks, ICLR 2025

---

> ### Author Response · Authors · 2025-03-17
>
> Thanks for pointing out those important issues. We have revised our work accordingly.
>
> Requested Changes 1: We have added more discussions in the Conclusion and Limitations part.
>
> Requested Changes 2: It is a typo. Thank you for your notice. We have corrected it.
>
> We thank you again for your valuable advice.

---

> > ### Comment · Reviewer_zqB3 · 2025-03-27
> >
> > Thanks for your response. I have read the added limitation part. I appreciate the theoretical results of the work, but I still have reservations about the practical application of this work. I believe this work can be further improved by making more contributions in the practical guidelines of how to design more effective ICL algorithms based on the theoretical findings.

---

> > > ### Author Response · Authors · 2025-04-01
> > >
> > > Thanks for your suggestion.
> > >
> > > We have to admit that most ICL papers are descriptive and explanatory rather than prescriptive. Our paper's theory part may provide a new perspective on the character of window size/context length. Currently, the major concern of restricting the window size has been computational due to the quadratic growth of the attention mechanism with respect to the context length, while our theory shows that limiting context length/window sizes can also statistically benefit the generalization if the data size is limited. We have modified our Conclusion and Limitations part accordingly.
> > >
> > > Thank you again for your time and advice.

---

### Review · Reviewer_5xWM · 2025-03-04

**Summary Of Contributions:**

This work analyzes the in-context learning capability of Transformers on both in-distribution and out-of-distribution regression tasks. Specifically, for in-distribution regression tasks, the authors present a generalization bound that exhibits a sharper decay rate using the window size $S$ and in-context size $T$. Empirically, they demonstrate that the in-context learning capability of a trained Transformer achieves a near-optimal risk comparable to that of the Bayes-optimal predictor.

For out-of-distribution regression tasks, the authors note that existing analyses of in-context learning have been established only for in-distribution settings, and thus investigate its effectiveness on out-of-distribution tasks. Empirically, they confirm that when appropriate training strategies are applied based on the specific out-of-distribution task such as task-shift, covariate-shift, and length-shift, the in-context learning capability of Transformers remains valid.

**Audience:**

Yes

**Broader Impact Concerns:**

This work investigates the in-context learning of the transformer and does not seem to raise the concerns of ethical issue.

**Claims And Evidence:**

Yes

**Requested Changes:**

* I recommend explaining the motivation behind the meta-distribution approach in Section 4.2 and reporting the corresponding results in the main section. This result appears to be one of the key findings, demonstrating that in-context learning with the appropriate training trick can be applied to covariate shift datasets.

*  I recommend providing a clearer explanation of Theorem 5.3 from Wu et al. (2023), described in last paragraph in section 4.3, to better support the justification of context length generalization. In the current draft, it is unclear how the absence of position encoding for Transformer, which improves context length generalization, relates to Theorem 5.3. If this connection cannot be adequately explained, it may be better to omit the use of Theorem 5.3 from Wu et al. (2023).

**Strengths And Weaknesses:**

### Strengths

*  This work presents a new shaper generalization bound of the in-context learning for in-distribution regression task.

*  This work investigates the capability of the in-context learning on out-of-distribution task revealing that proper training tricks is additionally necessary for the transformer to perform in-context learning capability on out-of-distribution tasks.


### Weaknesses

*  The claim of the in-context learning capability on out-of-distribution task seems less solid because the claims on out-of-distribution task are only empirically demonstrated and the necessary training tricks for Transformer is different depending on the types of out-of-distribution task. On the other hand, the claims on in-distribution tasks are supported by empirical results and theoretical analysis.

* Some parts of Chapter 4 are not clearly explained. For example, in Section 4.2 on covariate shift, the meta-distribution approach is introduced abruptly, making it unclear why it is considered. Additionally, the results of Section 4.2 are reported in the appendix, even though they could potentially be included in the main section. Furthermore, in the last paragraph of Section 4.3, the justification of context length generalization using Theorem 5.3 from Wu et al. (2023) is unclear, as the theorem is introduced without sufficient explanation of its relevance in this context.

* The experiments in this work are conducted only on regression tasks, raising doubts about the generalizability of the claims to classification tasks.

---

> ### Author Response · Authors · 2025-03-17
>
> Thanks for your valuable advice. We have revised our draft accordingly.
>
> Requested Change 1: We have moved that part from the appendix to the main text (Section 4.2). We've also added some intuitions on why we are considering such a training paradigm at the beginning of Section 4.2.
>
> Requested Change 2: We have added more discussions in the last paragraph of Section 4.3.
>
> We thank the reviewer again for all the suggestions.

---

> > ### Comment · Reviewer_5xWM · 2025-04-01
> >
> > Thanks for response on my review. I have confirmed how section 4.2 and 4.3 change.
> >
> > In section 4.2, I now better understand the intention behind the meta-distribution approach, though it is considered a crude motivation or observation.
> >
> > In Section 4.3, I do not understand how Theorem 5.3 from Wu et al. (2023) supports the claim made in this section. Based on my understanding, the empirical results in Section 4.3 suggest that the generalization of context length could be improved by not using position encoding. In contrast, Theorem 5.3 from Wu et al. (2023) states that generalization for context length works well when the training and test context lengths are similar. I find it difficult to see how these two seemingly different claims can be reconciled into a single coherent argument for Section 4.3.

---

> > > ### Author Response · Authors · 2025-04-01
> > >
> > > Thanks for your reply.
> > >
> > > Apologies for the confusion again. We use the claim from Wu et al. (2023) here because their model is also a simplified Transformer that skipped the position encoding part. Another point is that our empirical results also show that as the training and test context lengths get more different, the generalization becomes worse; that's of the same spirit as Wu et al. (2023).
> > >
> > > I hope this can clarify the point we made.

---

### Decision · Action_Editor_9dhL · 2025-04-02

**Recommendation:** Accept as is

**Comment:**

The AE is not an expert in this area, so relies on the judgment of the carefully selected expert reviewers.  The AE's assessment is that the reviewers performed careful reviews and provide balanced feedback on the strengths and weaknesses of the paper, as well as following up with author responses.

In final recommendations, the reviewers agree that the paper provides useful analysis and insights about downstream performance of in context learning and would be of particular interest to the UQ community.  Two of the reviewers note limitations in the thoroughness/rigor of theoretical analysis and potential for practical impact.  These opinions are elaborated in the reviews and comments that are viewable by the author.

Based on the comments and the recommendations of reviewers, the AE feels comfortable to recommend the paper for acceptance to TMLR.   The "as is" is meant to indicate that changes indicated in author responses should be incorporated, but additional significant changes are not required.  The AE does not recommend ICLR journal-to-conference track, in accordance with the majority recommendation of reviewers.

**Audience:**

Yes, all reviewers agree.

**Claims And Evidence:**

Yes, all reviewers agree.